# Gaussian Mixture Counterfactual Generator

**Jong-Hoon Ahn and Akshay Vashist**
Otsuka Pharmaceutical Development & Commercialization, Inc.
508 Carnegie Center Dr, Princeton, NJ 08540
{Jong-Hoon.Ahn, Akshay.Vashist}@otsuka-us.com

## Abstract

We address the individualized treatment effect (ITE) estimation problem, focusing on continuous, multidimensional, and time-dependent treatments for precision medicine. The central challenge lies in modeling these complex treatment scenarios while capturing dynamic patient responses and minimizing reliance on control data. We propose the Gaussian Mixture Counterfactual Generator (GMCG), a generative model that transforms the Gaussian mixture model—traditionally a tool for clustering and density estimation—into a new tool explicitly geared toward causal inference. This approach generates robust counterfactuals by effectively handling continuous and multidimensional treatment spaces. We evaluate GMCG on synthetic crossover trial data and simulated datasets, demonstrating its superior performance over existing methods, particularly in scenarios with limited control data. GMCG derives its effectiveness from modeling the joint distribution of covariates, treatments, and outcomes using a latent state vector while employing a conditional distribution of the state vector to suppress confounding and isolate treatment-outcome relationships.

## 1 Introduction

The advent of precision medicine is fundamentally reshaping clinical research, spotlighting the development of individualized treatment strategies tailored to each patient's distinct characteristics (Collins & Varmus, 2015; Jameson & Longo, 2015). This evolution has heightened the demand for sophisticated methods to estimate treatment effects, a process complicated by the intricate nature of real-world patient data. For instance, clinicians frequently encounter the challenge of optimizing multidimensional treatment regimens—where multiple therapeutic interventions are combined to achieve synergistic effects (Chen et al., 2019). Equally critical is the task of determining precise dosage combinations, which often involve continuous adjustments to maximize efficacy while minimizing adverse outcomes (Bica et al., 2020b). Moreover, patient responses are rarely static; they evolve over time due to physiological changes, disease progression, or external factors (Xu et al., 2016). These complexities underscore the need for tools capable of addressing modern healthcare applications' dynamic, multifaceted demands, such as personalized therapy design, clinical decision support systems, and patient outcome forecasting.

Despite notable advances, many existing approaches struggle to tackle these challenges in a unified manner. Some methods excel at modeling interactions among multiple treatments (Wang & Blei, 2019), while others focus on refining dosage optimization (Schwab et al., 2020) or capturing temporal dynamics in patient trajectories (Schulam & Saria, 2017). For instance, methods like propensity score matching are limited to fixed treatment categories, restricting their applicability in continuous dosage scenarios (Nagalapatti et al., 2024). However, these solutions often operate in isolation, lacking the flexibility to integrate the full spectrum of treatment variables—combinations, dosages, and time-dependent effects—within a single framework (Bica et al., 2021). This fragmentation poses a significant limitation, particularly in clinical scenarios where decisions must simultaneously account for these factors interplay. Additionally, traditional techniques frequently impose simplifying assumptions, such as fixed treatment categories or static patient states, failing to reflect real-world care's continuous and adaptive nature. Consequently, a critical gap remains for an approach that can cohesively address these interwoven dimensions while remaining adaptable to diverse data environments.

To address this, we introduce the Gaussian Mixture Counterfactual Generator (GMCG), a novel probabilistic model rooted in the foundational principles of counterfactual generation. GMCG employs a Gaussian mixture model (GMM) to capture the joint distribution of covariates, treatments, and outcomes, generating counterfactuals through a probability-driven approach that adheres rigorously to the core tenets of causal inference. By grounding its methodology in these established principles, GMCG ensures valid and interpretable estimation of individualized treatment effects (ITE) across complex scenarios, including those involving multidimensional treatments and continuous dosage spaces.

This paper makes three significant contributions to the field. First, it presents GMCG as a theoretical advancement that unifies previously disparate elements of treatment effect estimation, repurposing GMM—originally used for clustering or density estimation—into a robust framework for causal inference. Second, it offers a practical methodology for designing and evaluating personalized treatment strategies across varied clinical settings. Third, we validate GMCG's performance through extensive experiments on synthetic and simulated datasets, benchmarking it against state-of-the-art techniques. The paper is structured as follows: Section 2 reviews related work and identifies gaps in current approaches; Section 3 details the GMCG framework, modeling treatments as observed data and addressing the key challenge—counterfactual generation and ITE computation—with specific methods and formulas provided in Section 4; Section 5 presents experimental results and analysis; and Section 6 concludes with implications for causal inference, precision medicine, and future research. The appendices present the GMCG EM algorithm and pseudocode, among other details, to facilitate a deeper understanding of its implementation. Through this work, we aim to provide researchers and practitioners with a versatile tool to meet the evolving demands of personalized healthcare.

## 2 RELATED WORK

In common types of meta-learner, two distinct concepts are prevalent: the S-learner and the T-learner (Künzel et al., 2019; Okasa, 2022). The term 'S-learner' is derived from the notion of a single base learner, while 'T-learner' is an abbreviation for two base learners (Shalit et al., 2017). In a straightforward scenario involving a treatment group and a control group within a patient data set, the T-learner estimates two separate base learners: one for the treatment group data and another for the control group data. Subsequently, it computes the difference between these two base learners. Estimating ITEs based on Gaussian mixture models in the paper by (Ahn & Vashist, 2024) exemplifies a T-learner in counterfactual generation. The GMM-based counterfactual generator demonstrated superior performance compared to the synthetic control method on simulated low-density lipoprotein (LDL) data (Qian et al., 2021) and showed robustness even when dealing with heavily biased datasets.

However, the algorithm fundamentally operates as a treatment-controlled method, requiring a learned data distribution model on control units to generate counterfactual outcomes. This reliance poses limitations in scenarios where control data are unavailable, such as in crossover trials with no extended control group (Zhou et al., 2024). Additionally, its application as a T-learner is confined to data sets with a small, finite number of groups. As the number of groups increases, preparing distinct base learners, each trained on different group data becomes necessary. A significant limitation of such a T-learner is its inability to accommodate more complex real-world data, where potential treatments can span a multivariate, time-varying, and continuous domain.

Despite this limitation, the algorithm provides an important conceptual foundation for our work. Specifically, it functions as a T-learner, assuming separate distribution models for treatment and control data. Adapting the model into an S-learner framework allows us to develop a unified model capable of counterfactual inference even for unobserved treatment types. This adaptation allows us to generate counterfactual data even without relying on control data for training. Therefore, our proposed method builds upon the foundational idea, extending its applicability to complex treatment scenarios.

## 3 MODEL

In this section, we outline the latent variable model known as static state analysis (SSA), originally used in the paper by (Ahn & Vashist, 2024), and present how our new ideas are incorporated into this model. Building on the methodologies, we introduce two key problems. The first problem involves a brief overview of how to train the SSA model. The second problem concerns what counterfactual predictions to generate using the trained model.

### 3.1 STATIC STATE ANALYSIS (SSA)

The static state analysis decomposes a longitudinal dataset $\{\boldsymbol{x}^{(t)}\}$ into a time-varying observer matrix $\boldsymbol{W}^{(t)}$ and time-independent state vectors $\boldsymbol{s}$ with additive noises $\boldsymbol{\eta}^{(t)} \sim \mathcal{N}(\boldsymbol{0}, \boldsymbol{\Psi}^{(t)})$:

$$\boldsymbol{x}^{(t)} = \boldsymbol{W}^{(t)}\boldsymbol{s} + \boldsymbol{\eta}^{(t)}. \tag{1}$$

The time-independent state $\boldsymbol{s}$ is referred to as a static state. In this paper, longitudinal data $\boldsymbol{x}^{(t)}$ generally concatenate all different types of observed data, including baseline data $\boldsymbol{v}$ (e.g., demographic and genetic information), time-dependent covariates $\boldsymbol{l}^{(t)}$, results $\boldsymbol{y}^{(t)}$, and treatments $\boldsymbol{a}^{(t)}$. $\boldsymbol{x}^{(t)}$ and $\boldsymbol{W}^{(t)}$ can be written as $\boldsymbol{x}^{(t)} = [\boldsymbol{a}^{(t)}; \boldsymbol{l}^{(t)}; \boldsymbol{y}^{(t)}]$ and $\boldsymbol{W}^{(t)} = [\boldsymbol{W}_a^{(t)}; \boldsymbol{W}_l^{(t)}; \boldsymbol{W}_y^{(t)}]$ for $t \geq 1$. Equation (1) is decomposed into

$$\boldsymbol{a}^{(t)} = \boldsymbol{W}_a^{(t)}\boldsymbol{s} + \boldsymbol{\eta}_a^{(t)}, \tag{2}$$

$\boldsymbol{l}^{(t)} = \boldsymbol{W}_l^{(t)}\boldsymbol{s} + \boldsymbol{\eta}_l^{(t)}$, and $\boldsymbol{y}^{(t)} = \boldsymbol{W}_y^{(t)}\boldsymbol{s} + \boldsymbol{\eta}_y^{(t)}$ with $\boldsymbol{x}^{(0)} \equiv \boldsymbol{v} = \boldsymbol{W}_v^{(t)}\boldsymbol{s} + \boldsymbol{\eta}_v^{(t)}$. Equation (2) is our new equation that is directly included as observational data, whereas the original SSA model did not include it. Although it may appear to be a mere addition of a simple formula, it is important to note that this single addition fundamentally alters the nature and scope of the algorithm. Equations (1) and (2) also represent a factor model where the noise model follows a diagonal noise covariance matrix. This paper is not the first to model treatments or causes directly using a factor model. Wang & Blei (2019) and Bica et al. (2020a) successfully removed hidden confounding factors by directly modeling multiple causes as latent variables.

### 3.2 PROBLEM I: FITTING OF A MODEL TO A DATA SET

Given an observed data set $\mathbb{D} = \{\boldsymbol{x}_n^{(t)} | t \in \mathbb{T}_n, n = 1, \cdots, N\} \subset \mathcal{D}$, our first problem is to perform SSA by decomposing the data set $\mathbb{D}$ into a time-varying matrix $\boldsymbol{W}^{(t)}$ and a set of static state vectors $\mathbb{S} = \{\boldsymbol{s}_n | n = 1, \cdots, N\} \subset \mathcal{S}$. To learn the static state space $\mathcal{S}$ from the dataset, we employ a Gaussian mixture model (GMM) probabilistic distribution to model the static state as a vector-valued random variable, which is given by

$$p(\mathbf{s}) = \sum_{k=1}^{K} \pi_k \mathcal{N}(\mathbf{s}; \boldsymbol{\mu}_k, \boldsymbol{\Sigma}_k) \tag{3}$$

where $K$ is the number of mixture components. There is no more dependence on treatment notation because treatment data are directly included in the observational data. The problem is to estimate a set of parameters $\mathbb{Q} \equiv \{\{\pi_k\}, \{\boldsymbol{\mu}_k\}, \{\boldsymbol{\Sigma}_k\}, \boldsymbol{W}, \boldsymbol{\Psi}\}$ that maximizes the expected complete-data log-likelihood by using the expectation and maximization algorithm:

$$\hat{\mathbb{Q}} = \arg\max_{\mathbb{Q}} \sum_{n=1}^{N} \sum_{t \in \mathbb{T}_n} \langle \ln p(\boldsymbol{x}_n^{(t)}, \mathbf{s}_n | \mathbb{Q}) \rangle \tag{4}$$

where $\langle \mathbf{s}_n \rangle = \int |d\mathbf{s}| p(\mathbf{s}|\boldsymbol{x}_n, \mathbb{Q}) \times \mathbf{s}$ and $\langle \mathbf{s}_n \mathbf{s}_n^T \rangle = \int |d\mathbf{s}| p(\mathbf{s}|\boldsymbol{x}_n, \mathbb{Q}) \times \mathbf{s}\mathbf{s}^T$. Then, we can also get the noise vector as $\langle \boldsymbol{\eta}_n^{(t)} \rangle = \boldsymbol{x}_n^{(t)} - \widehat{\boldsymbol{W}}^{(t)} \langle \mathbf{s}_n \rangle$. Through GMM, latent states $\boldsymbol{s}$ can reflect various clusters (*e.g.*, patient types, conditions), which is essential for predicting various potential outcomes in counterfactual scenarios.

### 3.3 PROBLEM II: COUNTERFACTUAL GENERATION WITH NO BACKWARD CAUSATION

Let $\hat{\mathbb{Q}}$ be a maximizer of the log-likelihood function by Equation (4) for a factual data set $\mathbb{D}_f = \{\boldsymbol{x}_{n,f}^{(:)} | n = 1, \cdots, N\}$. The main problem in our paper is to generate a synthetic counterfactual

longitudinal dataset $\mathbb{D}_{\text{cf}}^{(\tau+\delta:)} = \{x_{n,\text{cf}}^{(\tau+\delta:)}|n = 1, \cdots, N\}$ from using $\hat{\mathbb{Q}}$ when we assume that we had assigned alternative treatments $a_{n,\text{cf}}^{(\tau:\tau+\delta)}$ instead of $a_{n,\text{f}}^{(\tau:\tau+\delta)}$ where we define the colon notation as concatenating time-varying data or parameters:

$$x \equiv x^{(:)} \equiv \begin{bmatrix} x^{(0)} \\ x^{(1)} \\ \vdots \end{bmatrix}, \quad W^{(:\tau)} \equiv W^{(0:\tau)} \equiv \begin{bmatrix} W^{(0)} \\ \vdots \\ W^{(\tau-1)} \end{bmatrix}, \quad a^{(\tau:\tau+\delta)} \equiv \begin{bmatrix} a^{(\tau)} \\ \vdots \\ a^{(\tau+\delta-1)} \end{bmatrix}. \quad (5)$$

Counterfactual pretreatment data must align as closely as possible with the provided factual pretreatment data (Abadie & Gardeazabal, 2003; Doudchenko et al., 2021). Thus, equivalently, the above problem can be written as generating an entire dataset $\mathbb{D}_{\text{cf}} = \{x_{n,\text{cf}}^{(:)}|n = 1, \cdots, N\}$ that satisfies $x_{n,\text{cf}}^{(:\tau)} = x_{n,\text{f}}^{(:\tau)}$.

The subscript "f" stands for factual, and "cf" is derived from the first letters of counter-factual. Data obtained through observation is called factual data. On the other hand, data generated through counterfactual reasoning without being observed in the real world is referred to as counterfactual data in this paper. Counterfactual reasoning involves considering hypothetical scenarios, such as imagining what would happen if a patient who took medication at time $\tau$ had not taken it. Before time $\tau$, the patient actually did not take the medication, so the factual data and counterfactual data are the same, which can be expressed as $x_{n,\text{cf}}^{(:\tau)} = x_{n,\text{f}}^{(:\tau)}$. The goal of this paper is to generate counterfactual data. This can be interpreted in the context of clinical trials as virtual patient data or a synthetic placebo arm. In the next section, we introduce the basic assumptions of counterfactual reasoning and propose a new method for counterfactual prediction.

## 4 METHODS

To solve the second problem, this section details the specific assumptions and formulas for generating counterfactual predictions.

### 4.1 ASSUMPTIONS

A probabilistic observation refers to a process of observing outcomes that are not deterministic but rather governed by probability distributions. In other words, instead of a single fixed outcome, there are multiple possible outcomes, each associated with a certain probability of occurrence. In this paper, probabilistic observation is formalized using the SSA model as follows:

---

**Probabilistic Observation**

Consider a patient with pretreatment data $x_{\text{f}}^{(:\tau)}$, who receives a course of treatment $a_{\text{f}}^{(\tau:\tau+\delta)}$. Post-treatment data are obtained by monitoring the patient's state vector $s_{\text{f}}$ using observers $W^{(\tau:)}$ and additive noises $\eta^{(\tau:)}$:

$$x_{\text{f}}^{(\tau:)} = W^{(\tau:)} s_{\text{f}} + \eta^{(\tau:)} \quad (6)$$

where $s_{\text{f}}$ is a realization of

$$s_{\text{f}} \sim p(s|x_{\text{f}}^{(:\tau)}, a_{\text{f}}^{(\tau:\tau+\delta)}). \quad (7)$$

---

This formulation implies a probabilistic observation process in which future outcomes are probabilistically determined. However, the random generation of a state vector from Equation (7) must not affect the past information prior to $\tau$ or the treatment information. Thus, we have:

$$\begin{bmatrix} x_{\text{f}}^{(:\tau)} \\ a_{\text{f}}^{(\tau:\tau+\delta)} \end{bmatrix} = \begin{bmatrix} W^{(:\tau)} \\ W_a^{(\tau:\tau+\delta)} \end{bmatrix} s_{\text{f}} + \begin{bmatrix} \eta^{(:\tau)} \\ \eta_a^{(\tau:\tau+\delta)} \end{bmatrix}. \quad (8)$$

The process of probabilistic observation is not limited exclusively to the acquisition of factual data. When an alternative treatment, distinct from the one administered, is posited, this same process can be applied to generate counterfactual data. This is restated as counterfactual generation as follows:

> **Counterfactual Generation**
>
> Consider a patient with pretreatment data $\boldsymbol{x}_{\mathrm{f}}^{(:\tau)}$, who, instead of receiving the actual treatment $\boldsymbol{a}_{\mathrm{f}}^{(\tau:\tau+\delta)}$, is assumed to have received an alternative treatment $\boldsymbol{a}_{\mathrm{cf}}^{(\tau:\tau+\delta)}$. Post-treatment data would be obtained by monitoring the patient's state vector $\mathbf{s}_{\mathrm{cf}}$ using the same observers $\boldsymbol{W}^{(\tau:)}$ and noises $\boldsymbol{\eta}^{(\tau:)}$:
>
> $$\boldsymbol{x}_{\mathrm{cf}}^{(\tau:)} = \boldsymbol{W}^{(\tau:)}\boldsymbol{s}_{\mathrm{cf}} + \boldsymbol{\eta}^{(\tau:)} \tag{9}$$
>
> where $\boldsymbol{s}_{\mathrm{cf}}$ is a realization of
>
> $$\mathbf{s}_{\mathrm{cf}} \sim p(\mathbf{s}|\boldsymbol{x}_{\mathrm{f}}^{(:\tau)}, \boldsymbol{a}_{\mathrm{cf}}^{(\tau:\tau+\delta)}). \tag{10}$$

This formulation also implies that the random generation of a counterfactual state vector from Equation (10) must not affect the past information prior to $\tau$ or the treatment information. Thus, we have:

$$\begin{bmatrix} \boldsymbol{x}_{\mathrm{f}}^{(:\tau)} \\ \boldsymbol{a}_{\mathrm{cf}}^{(\tau:\tau+\delta)} \end{bmatrix} = \begin{bmatrix} \boldsymbol{W}^{(:\tau)} \\ \boldsymbol{W}_a^{(\tau:\tau+\delta)} \end{bmatrix} \boldsymbol{s}_{\mathrm{cf}} + \begin{bmatrix} \boldsymbol{\eta}^{(:\tau)} \\ \boldsymbol{\eta}_a^{(\tau:\tau+\delta)} \end{bmatrix}. \tag{11}$$

where $\boldsymbol{\eta}$ is the same noise as in the factual data observation from Equations (6) and (8). If the patient were to return exactly to the time $\tau$ and receive a different treatment, only the patient's state vector would change due to the altered treatment, while the additive noise, acting independently of the changed treatment, should remain the same.

The counterfactual generation described above assumes a retrospective perspective. It refers to imagining alternative outcomes to past events, essentially considering what could have been if different actions or circumstances had occurred. The current time is $t > \tau + \delta$. However, Equations (9) and (10) can still be applied from a prospective perspective. That is, it can be written as *"If the patient were to receive an alternative treatment, the post-treatment data would be obtained by Equations (9) and (10)."* In this case, the current time is $t = \tau$. If the retrospective perspective is valid for post-analysis (e.g. after completing a clinical trial), the prospective perspective can be valid for ongoing decision making (e.g. virtual twins).

## 4.2 Generating a Counterfactual State

Once we find a maximizer $\hat{\mathbb{Q}}$ estimated by solving problem I using EM algorithms (see Appendix B), we can sample a factual state $\boldsymbol{s}_{\mathrm{f}}$ from $p(\mathbf{s}|\boldsymbol{x}_{\mathrm{f}}^{(:\tau)}, \boldsymbol{a}_{\mathrm{f}}^{(\tau:\tau+\delta)})$ or obtain its expectation $\boldsymbol{s}_{\mathrm{f}} \equiv \langle\mathbf{s}_{\mathrm{f}}\rangle$ for a new patient with data $\boldsymbol{x}_{\mathrm{f}}^{(:\tau)}$ who took the treatment $\boldsymbol{a}_{\mathrm{f}}^{(\tau:\tau+\delta)}$. As the first step in counterfactual data generation for the patient, we can generate a counterfactual state by realizing a vector $\boldsymbol{s}_{\mathrm{cf}}$ from Equation (3)

$$\mathbf{s}_{\mathrm{cf}} \sim \sum_{k=1}^{K} \pi_k \mathcal{N}(\mathbf{s}; \boldsymbol{\mu}_k, \boldsymbol{\Sigma}_k) \tag{12}$$

that is conditioned on, by subtracting Equation (11) from Equation (8),

$$\begin{bmatrix} \mathbf{0} \\ \boldsymbol{a}_{\mathrm{cf}}^{(\tau:\tau+\delta)} - \boldsymbol{a}_{\mathrm{f}}^{(\tau:\tau+\delta)} \end{bmatrix} = \begin{bmatrix} \boldsymbol{W}^{(:\tau)} \\ \boldsymbol{W}_a^{(\tau:\tau+\delta)} \end{bmatrix} (\boldsymbol{s}_{\mathrm{cf}} - \boldsymbol{s}_{\mathrm{f}}). \tag{13}$$

Equivalently, we can write a counterfactual state random vector in the form of

$$\begin{aligned} \boldsymbol{s}_{\mathrm{cf}} &= \boldsymbol{s}_{\mathrm{f}} + \begin{bmatrix} \boldsymbol{W}^{(:\tau)} \\ \boldsymbol{W}_a^{(\tau:\tau+\delta)} \end{bmatrix}^{\dagger} \begin{bmatrix} \mathbf{0} \\ \boldsymbol{a}_{\mathrm{cf}}^{(\tau:\tau+\delta)} - \boldsymbol{a}_{\mathrm{f}}^{(\tau:\tau+\delta)} \end{bmatrix} + \ker\left(\begin{bmatrix} \boldsymbol{W}^{(:\tau)} \\ \boldsymbol{W}_a^{(\tau:\tau+\delta)} \end{bmatrix}\right) \boldsymbol{\xi} \\ &\equiv \boldsymbol{s}_{\mathrm{f}} + \boldsymbol{\delta}_a + \boldsymbol{N}\boldsymbol{\xi} \end{aligned} \tag{14}$$

where dagger and ker() denote the pseudo-inverse and a basis of the kernel of $[\boldsymbol{W}^{(:\tau)}; \boldsymbol{W}_a^{(\tau:\tau+\delta)}]$, respectively, and $\boldsymbol{\xi}$ is a random vector of the following distribution by denoting a state shift by treatment change as $\boldsymbol{\delta}_a$ and the kernel matrix as $\boldsymbol{N}$:

$$\boldsymbol{\xi} \sim \sum_{k=1}^{K} p_k \mathcal{N}(\boldsymbol{\xi}; \boldsymbol{m}_k, \boldsymbol{C}_k) \tag{15}$$

with

$$C_k = (N^T \Sigma_k^{-1} N)^{-1} \tag{16}$$

$$m_k = C_k N^T \Sigma_k^{-1}(\mu_k - s_f - \delta_a) \tag{17}$$

$$p_k = \frac{\pi_k \mathcal{N}(N m_k + s_f + \delta_a | \mu_k, \Sigma_k)|C_k|^{\frac{1}{2}}}{\sum_{k'=1}^{K} \pi_{k'} \mathcal{N}(N m_{k'} + s_f + \delta_a | \mu_{k'}, \Sigma_{k'})|C_{k'}|^{\frac{1}{2}}}. \tag{18}$$

Notice that $\delta_a$ in Equations (14), (17), and (18) is the new term derived by introducing Equation (2), and makes a crucial contribution to this paper.

## 4.3 GENERATING COUNTERFACTUAL OUTCOMES

From Equations (6), (9), and (14), we can generate a counterfactual data vector based on the Gaussian mixture: $x_{cf} = x_f + W(\delta_a + N\xi)$ where $\xi$ is a realization of Equation (15). More formally, we can state from a retrospective perspective:

---

**Gaussian Mixture Counterfactual Generator**

If a patient with factual data $x_f$ had received an alternative treatment $a_{cf}^{(\tau:\tau+\delta)}$, the counterfactual data that we would have obtained is represented as a random variable given by

$$\mathbf{x}_{cf} \sim \sum_{k=1}^{K} p_k \mathcal{N}\left(\mathbf{x}; x_f + W(\delta_a + N m_k), W N C_k N^T W^T\right) \tag{19}$$

where $p_k$, $m_k$, and $\delta_a$ are functions of $a_{cf}^{(\tau:\tau+\delta)}$ as given by Equations (14), (17), and (18).

---

This formulation reflects a probabilistic approach for generating counterfactual outcomes based on a *single* Gaussian mixture model. Note that the GMCG presented in Equation (19) is not a statically fixed GMM model but a complex model where $p_k$, $\delta_a$, $N$, $C_k$, and $m_k$ change by $\tau$, $\delta$, and $a_{cf}$. Equation (19) is expressed in terms of complex mean vectors and covariance matrices, but it can be implemented more simply by separating it into a sampling process and a linear transformation.

Because $W^{(:\tau)}\delta_a = W^{(:\tau)}N = 0$, the random variable $\mathbf{x}_{cf}^{(:\tau)}$ is always the same as $x_f^{(:\tau)}$ by Equation (19). However, the difference in outcomes made by a particular vector $\xi$ in the probabilistic model in Equation (15) is a time-varying *individualized treatment effect* (ITE) caused by the difference in treatments $a_{cf}^{(\tau:\tau+\delta)} - a_f^{(\tau:\tau+\delta)}$ that we can obtain

$$\text{ITE} \equiv y_{cf}^{(\tau:)} - y_f^{(\tau:)} = W_y^{(\tau:)}(\delta_a + N\xi). \tag{20}$$

The expected ITE (eITE) we can obtain on average is

$$\mathbb{E}[\text{ITE}] \equiv \mathbb{E}[\mathbf{y}^{(\tau:)}|x_f^{(:\tau)}, a_{cf}^{(\tau:\tau+\delta)}] - y_f^{(\tau:)} = W_y^{(\tau:)}(\delta_a + N\sum_{k=1}^{K} p_k m_k). \tag{21}$$

By using Equations (15), (17), and (18), we can calculate the *conditional averaged treatment effect* (CATE) from Equation (20)

$$\begin{aligned} \text{CATE} &\equiv \mathbb{E}[\mathbf{y}^{(\tau+\delta:)}|x_f^{(:\tau)}, a_{cf}^{(\tau:\tau+\delta)}] - \mathbb{E}[\mathbf{y}^{(\tau+\delta:)}|x_f^{(:\tau)}, a_f^{(\tau:\tau+\delta)}] \\ &= W_y^{(\tau:)}N\sum_{k=1}^{K}(p_k m_k - q_k n_k) \end{aligned} \tag{22}$$

where

$$n_k = C_k N^T \Sigma_k^{-1}(\mu_k - s_f) \tag{23}$$

$$q_k = \frac{\pi_k \mathcal{N}(N n_k + s_f | \mu_k, \Sigma_k)|C_k|^{\frac{1}{2}}}{\sum_{k'=1}^{K} \pi_{k'} \mathcal{N}(N n_{k'} + s_f | \mu_{k'}, \Sigma_{k'})|C_{k'}|^{\frac{1}{2}}}. \tag{24}$$

## 5 EXPERIMENTS

### 5.1 ILLUSTRATIVE EXAMPLE OF A CROSSOVER TRIAL

To demonstrate the effectiveness of GMCG in handling continuous treatment spaces and time-dependent effects at the individual patient level, we present a crossover trial as an illustrative example. A crossover trial is a clinical study design in which participants receive different treatments in a sequential manner, offering a robust framework for evaluating the prolonged efficacy and safety of new therapeutic interventions (Elbourne et al., 2002; Nolan et al., 2016; Dwan et al., 2019). In this design, patients are initially randomized into two groups—one receiving the active treatment and the other a placebo—for a defined period. Following this, the placebo group transitions to the active treatment during an open-label extension phase. This approach ensures that all participants eventually receive the treatment, addressing ethical considerations while enabling the observation of sustained treatment responses. However, a key challenge arises in the extension phase due to the lack of concurrent control data, making it difficult to distinguish treatment effects from natural disease progression. By employing the GMCG, we address this issue by generating synthetic counterfactual data, capitalizing on its ability to model continuous variations in treatment and capture dynamic, time-dependent patient outcomes effectively.

This section demonstrates how the proposed GMCG algorithm by Equation (19) can be applied to crossover trial data to generate synthetic control data for illustrative purposes. The data set shown here is not actual but derived from a simulation model, the details of which can be found in Appendix C. In the left panel of Figure 1, data from two arms are depicted using shades of red and blue. The red-shaded arm represents groups that received the assigned doses over an entire 24-week period, while the blue-shaded arm represents groups that received the doses only during the last 12 weeks. Each arm is further subdivided into groups based on different doses of 1 mg and 2 mg, resulting in a total of four distinct dose groups. Notably, no group remains on placebo for the entire duration, necessitating the creation of a synthetic control arm to compare to the treatment arms to estimate the efficacy of the treatment. The GMCG model was trained on 400 patients' outcome data across the four groups, each consisting of 100 patients, without additional covariates.

The four thick lines of different colors on the left are randomly selected and displayed, and they are further dealt with in the four subplots on the right. The first subplot displays data from the 1mg group during the second 12-week period, with the algorithm-generated synthetic data marked by the red dashed line. Comparing these synthetic data with the observed data and the ground-truth control data, respectively, we observe a perfect match during the first 12 weeks and a close alignment in the second half. The difference between the observed 12-week 1mg data and the red dashed synthetic data after the 12th week can be interpreted as the patient's expected individual treatment effect (eITE) estimated by Equation (21). The critical point is that the synthetic control arm created through the proposed algorithm corresponds individually to the active or crossover arm,

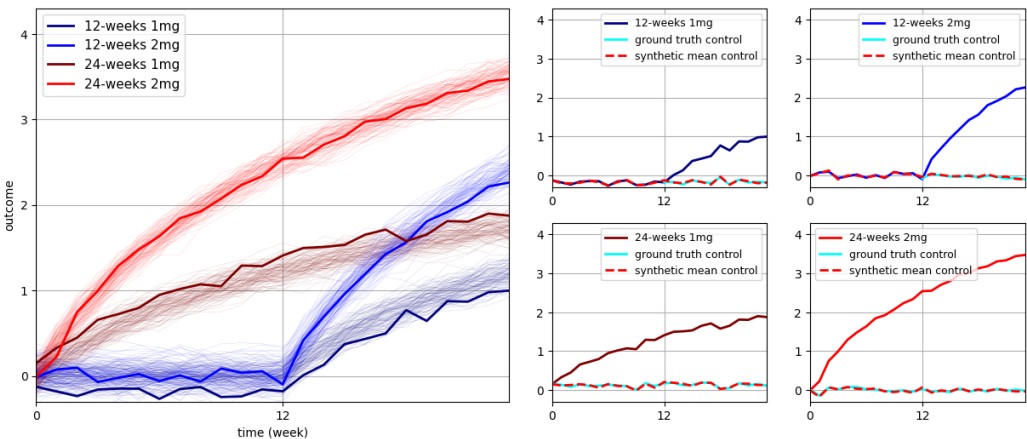

Figure 1: Illustrative crossover trial dataset (linear efficacy) and synthetic control data generation

forming a synthetic individually controlled arm. Generating and utilizing the synthetic individually controlled arm makes it possible to estimate ITEs or eITEs.

This simple example clearly illustrates a problem that does not have a readily available logical solution despite its apparent simplicity in the reference. For example, look at the blue-colored data. Seemingly, synthetic control methods could create synthetic control data for the data. However, traditional synthetic control methods rely on the availability of control units, which are absent here, to approximate it by taking the weighted sum of them. Thus, this algorithm provides an innovative way to infer these control data by circumventing the need for explicit control units, highlighting its capability to solve problems that traditional algorithms cannot address.

## 5.2 SIMULATED DATA

### 5.2.1 LDL CHOLESTEROL DATA

For quantitative analysis, LDL cholesterol simulated data was utilized. Data for the treatment and control groups were created using the PK/PD model (Faltaos et al., 2006; Yokote et al., 2008; Kim et al., 2011) of statin drugs prescribed to hypercholesterolaemic patients. The experiment was conducted using data with the same characteristics as those generated by SyncTwin (Qian et al., 2021) and SSA-GMM (Ahn & Vashist, 2024).

As explained in Section 2 on the motivation of this paper, the proposed algorithm was developed by converting SSA-GMM into the S-learner framework. While this greatly enhances its versatility by allowing application to various types of treatments, its performance should remain equivalent to the original GMM-based counterfactual generator. However, we achieved better results by finding optimized parameters in the GMCG algorithm. As shown in Table 1 below, the MAE of SSA-GMM reached a level of 0.07, but GMCG was able to reduce the error to a much lower level. When compared to the SyncTwin, SCM (Synthetic Control Method), and CRN (Bica et al., 2020b) models, the error gap becomes even larger.

Table 1: LDL cholesterol data experiment results

| Method | $N = N_0 + N_1 = 200 + 200 = 400$ | | | $N = N_0 + N_1 = 1000 + 200 = 1200$ | | |
|---|---|---|---|---|---|---|
| | $p_0 = 0.1$ | $p_0 = 0.25$ | $p_0 = 0.5$ | $p_0 = 0.1$ | $p_0 = 0.25$ | $p_0 = 0.5$ |
| GMCG | **0.042** | **0.038** | **0.043** | **0.040** | **0.039** | **0.035** |
| SSA-GMM | 0.072 | 0.072 | 0.076 | 0.070 | 0.073 | 0.069 |
| SyncTwin | 0.308 | 0.150 | 0.116 | 0.178 | 0.106 | 0.094 |
| SCM | 0.341 | 0.151 | 0.150 | 0.231 | 0.172 | 0.158 |
| CRN | 0.530 | 0.631 | 0.343 | 0.456 | 0.404 | 0.371 |

### 5.2.2 LDL CHOLESTEROL DATA WITH NO CONTROLS

The reason for using LDL cholesterol data is not just to show superiority over SSA-GMM and the other algorithms. As sketched in the introduction and illustratively shown in Section 5.1, it is intended to be used for more quantitative analysis of crossover trials. However, the composition of the data differs from that in Table 1. In Table 1, $N_0 = 200$ control data and $N_1 = 200$ treatment data were used. In addition, $N_0 = 1000$ control data and $N_1 = 200$ treatment data were also used as training data. But now, we aim to evaluate a much more challenging task: using no control data at all. This is logically unsound because comparing the results of the treatment data with the control data is the golden standard for determining the treatment effect. Without comparison, the effect of the treatment cannot be discussed. Instead, we assumed that we would test the experimental data at various doses. The total duration of the experiment is $T = 30$. Until time $t = 25$, no patient receives the drug. After time $t = 25$, each patient takes a random dose of statin between 0mg and 10mg. This marks a crossover point at time $t = 25$.

After training the model using random doses data, we tested it on the 10mg test data used in Table 1 and obtained the results shown in Table 2. When comparing the results for $N = 1200$, the performance was similar to SCM. Increasing $N$ slightly improved the results. However, the data for $N = 400$ did not train properly. The results in Table 1 were obtained using control data, while

the results in Table 2 were obtained by training the model without control data, highlighting the notable performance of GMCG. As explained in the Related Work section, models like SCM and SyncTwin do not function without control data because they create synthetic control by finding linear or nonlinear combinations of control units. Although recurrent network models like CRN can be tested, considering the poor results in Table 1, it is not possible to obtain proper results without training with control data. The results in Tables 1 and 2 are numerical results in terms of MAE, and examples for visual reference are provided in the Appendix D.

Table 2: LDL cholesterol data experiment results (no control group)

| Method | $N = 1200$ | | | $N = 2000$ | | |
|--------|------------|--|--|------------|--|--|
| | $p_0 = 0.1$ | $p_0 = 0.25$ | $p_0 = 0.5$ | $p_0 = 0.1$ | $p_0 = 0.25$ | $p_0 = 0.5$ |
| GMCG | 0.251 | 0.199 | 0.168 | 0.184 | 0.150 | 0.152 |

GMM-type algorithms use the EM algorithm to find the local maximum of the given likelihood function. The EM algorithm for GMCG training also finds the local maximum of the likelihood function (see Appendix B for the EM algorithm for GMCG). This means that results are influenced by the initial values. In the GMCG algorithm for crossover trials, the choice of initial values affected performance. Applying the EM algorithm from random values may make it difficult to achieve results like those in Table 2. The data structure of crossover trials is fundamentally curved and requires multiple mixture components. The synthetic control data by zero dose is particularly influenced by mixture components derived from low-dose data. Therefore, the GMCG-EM algorithm should be executed using values obtained by first applying a GMM to data created by low doses. The results in Table 2 used two mixture components for dose data between 0mg and 3mg, one for 3mg to 7mg, and one for 7mg to 10mg. Thus, a total of $K = 4$ mixture components were used.

Although the MAE is larger compared to the GMCG results in Table 1, this method has potentially powerful implications. First, to the best of the authors' knowledge, this is the first attempt to solve a causal inference problem without using control data. In the field of clinical trials, there are single-arm trials (Wang et al., 2024), but they make control groups using historical or external controls. However, the GMCG method structurally learns the dose response from the experimental data alone and extrapolates the prediction for zero dose. Second, clinical trials without any control data present a new form of clinical trials that are more economical, easier to recruit patients, and more ethically suitable for patients in many cases where historical or external controls cannot be sufficiently obtained, thus opening up broader possibilities.

## 6 CONCLUSION

This study introduces the Gaussian mixture counterfactual generator (GMCG) as a novel approach to estimating individualized treatment effects (ITE) in precision medicine. By adapting the Gaussian mixture model (GMM) into a robust framework for causal inference, GMCG addresses challenges that traditional methods struggle with, such as continuous treatment doses, multidimensional treatments, and time-varying effects. Its probabilistic nature allows it to effectively capture the complexity of real-world treatment scenarios, suggesting its potential to contribute meaningfully to developing personalized treatment strategies.

The experimental results lend support to GMCG's practical utility. Using synthetic crossover trial data, it reliably generated counterfactuals even without concurrent controls, and it outperformed state-of-the-art methods like SCM, SyncTwin, and CRN on simulated LDL cholesterol and tumor growth datasets. Notably, GMCG maintained strong performance despite limited control data, hinting at its applicability to real clinical settings for personalized treatment planning and decision support. Moving forward, validating GMCG with real-world clinical data and extending the model to handle diverse treatment types—while relaxing assumptions like Gaussian noise—could enhance its flexibility and broaden its impact.

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

# A  NOTATIONS

## Numbers and Arrays

| | |
|---|---|
| $a$ | A scalar (integer or real) |
| $\boldsymbol{a}$ | A vector |
| $\boldsymbol{A}$ | A matrix |
| $\boldsymbol{I}_n$ | Identity matrix with $n$ rows and $n$ columns |
| $\boldsymbol{I}$ | Identity matrix with dimensionality implied by context |
| $\mathrm{diag}(\boldsymbol{a})$ | A square, diagonal matrix with diagonal entries given by $\boldsymbol{a}$ |
| a | A scalar random variable |
| $\mathbf{a}$ | A vector-valued random variable |
| $\mathbf{A}$ | A matrix-valued random variable |
| $[\boldsymbol{a};\ \boldsymbol{a}] \equiv \begin{bmatrix} \boldsymbol{a} \\ \boldsymbol{a} \end{bmatrix}$ | Vectors concatenation |
| $[\boldsymbol{A};\ \boldsymbol{A}] \equiv \begin{bmatrix} \boldsymbol{A} \\ \boldsymbol{A} \end{bmatrix}$ | Matrices concatenation |

## Sets and Graphs

| | |
|---|---|
| $\mathbb{A}$ | A set |
| $\mathbb{R}$ | The set of real numbers |
| $\{0, 1\}$ | The set containing 0 and 1 |
| $\{0, 1, \ldots, n\}$ | The set of all integers between $0$ and $n$ |

## Probability and Information Theory

| | |
|---|---|
| $P(\mathrm{a})$ | A probability distribution over a discrete variable |
| $p(\mathrm{a})$ | A probability distribution over a continuous variable, or over a variable whose type has not been specified |
| $\mathrm{a} \sim P$ | Random variable a has distribution $P$ |
| $\mathbb{E}_{\mathrm{x} \sim P}[f(x)]$ or $\mathbb{E}f(x)$ | Expectation of $f(x)$ with respect to $P(\mathrm{x})$ |
| $\mathrm{Var}(f(x))$ | Variance of $f(x)$ under $P(\mathrm{x})$ |
| $\mathrm{Cov}(f(x), g(x))$ | Covariance of $f(x)$ and $g(x)$ under $P(\mathrm{x})$ |
| $\mathcal{N}(\boldsymbol{x}; \boldsymbol{\mu}, \boldsymbol{\Sigma})$ | Gaussian distribution over $\boldsymbol{x}$ with mean $\boldsymbol{\mu}$ and covariance $\boldsymbol{\Sigma}$ |

## B    EM ALGORITHMS FOR TRAINING GMCG

For the case of having noise $\boldsymbol{\eta}^{(t)} \sim \mathcal{N}(\mathbf{0}, \boldsymbol{\Psi}^{(t)})$ with a diagonal matrix $\boldsymbol{\Psi}^{(t)}$, Eq. (1) implies a probability distribution over $\boldsymbol{x}^{(t)}$-space for a given $\mathbf{s} \in \mathbb{R}^M$ of the form

$$p(\boldsymbol{x}^{(t)}|\boldsymbol{s}) = |2\pi\boldsymbol{\Psi}^{(t)}|^{-\frac{1}{2}} \exp\left\{-\frac{1}{2}(\boldsymbol{x}^{(t)} - \boldsymbol{W}^{(t)}\boldsymbol{s})^T(\boldsymbol{\Psi}^{(t)})^{-1}(\boldsymbol{x}^{(t)} - \boldsymbol{W}^{(t)}\boldsymbol{s})\right\}. \tag{25}$$

With a Gaussian mixture prior over $\boldsymbol{s}$ defined by Eq. (3), we obtain the marginal distribution of $\boldsymbol{x}^{(t)}$ in the form

$$p(\boldsymbol{x}^{(t)}) = \sum_{k=1}^{K} \pi_k \cdot |2\pi\boldsymbol{V}_k^{(t)}|^{-\frac{1}{2}} \exp\left\{-\frac{1}{2}(\boldsymbol{x}^{(t)} - \boldsymbol{W}^{(t)}\boldsymbol{\mu}_k)^T \boldsymbol{V}_k^{(t)-1}(\boldsymbol{x}^{(t)} - \boldsymbol{W}^{(t)}\boldsymbol{\mu}_k)\right\}, \tag{26}$$

where the model covariance is $\boldsymbol{V}_k^{(t)} = \boldsymbol{\Psi}^{(t)} + \boldsymbol{W}^{(t)}\boldsymbol{\Sigma}_k\boldsymbol{W}^{(t)T}$.

We can also use the colon notation $\boldsymbol{x} \equiv \boldsymbol{x}^{(:)}$ and $\boldsymbol{W} \equiv \boldsymbol{W}^{(:)}$ defined by Eq. (5). Then, we have the conditional distribution and the marginal distribution in the form of

$$p(\boldsymbol{x}|\mathbf{s}) = |2\pi\boldsymbol{\Psi}|^{-\frac{1}{2}} \exp\left\{-\frac{1}{2}(\boldsymbol{x} - \boldsymbol{W}\boldsymbol{s})^T\boldsymbol{\Psi}^{-1}(\boldsymbol{x} - \boldsymbol{W}\boldsymbol{s})\right\} \tag{27}$$

$$p(\boldsymbol{x}) = \sum_{k=1}^{K} \pi_k \cdot |2\pi\boldsymbol{V}_k|^{-\frac{1}{2}} \exp\left\{-\frac{1}{2}(\boldsymbol{x} - \boldsymbol{W}\boldsymbol{\mu}_k)^T\boldsymbol{V}_k^{-1}(\boldsymbol{x} - \boldsymbol{W}\boldsymbol{\mu}_k)\right\}. \tag{28}$$

where $\boldsymbol{V}_k = \boldsymbol{\Psi} + \boldsymbol{W}\boldsymbol{\Sigma}_k\boldsymbol{W}^T$. By Bayes' rule, this leads to the posterior distribution of the form

$$
\begin{aligned}
p(\boldsymbol{s}|\boldsymbol{x}) &= \sum_k p(\boldsymbol{s}|\boldsymbol{x}, k)p(k|\boldsymbol{x}) = \sum_k p(k|\boldsymbol{x}) \cdot p(\boldsymbol{x}|\boldsymbol{s})p(\boldsymbol{s}|k)/p(\boldsymbol{x}|k) \\
&= \sum_k p(k|\boldsymbol{x}) \cdot |2\pi\boldsymbol{M}_k|^{-\frac{1}{2}} \exp\Big\{-\frac{1}{2}\Big((\boldsymbol{s}-\boldsymbol{\mu}_k) - \boldsymbol{M}_k\boldsymbol{W}^T\boldsymbol{\Psi}^{-1}(\boldsymbol{x}-\boldsymbol{W}\boldsymbol{\mu}_k)\Big)^T \\
&\qquad\qquad \times \boldsymbol{M}_k^{-1}\Big((\boldsymbol{s}-\boldsymbol{\mu}_k) - \boldsymbol{M}_k\boldsymbol{W}^T\boldsymbol{\Psi}^{-1}(\boldsymbol{x}-\boldsymbol{W}\boldsymbol{\mu}_k)\Big)\Big\}
\end{aligned} \tag{29}
$$

where $\boldsymbol{M}_k = (\boldsymbol{\Sigma}_k^{-1} + \boldsymbol{W}^T\boldsymbol{\Psi}^{-1}\boldsymbol{W})^{-1}$.

Now we have the joint distribution of observational data $\boldsymbol{x}_n$ and the latent variables $\mathbf{s}_n$ in the form of

$$
\begin{aligned}
p(\boldsymbol{x}_n, \boldsymbol{s}_n) &= |2\pi\boldsymbol{\Psi}|^{-\frac{1}{2}} \exp\left\{-\frac{1}{2}(\boldsymbol{x}_n - \boldsymbol{W}\boldsymbol{s}_n)^T\boldsymbol{\Psi}^{-1}(\boldsymbol{x}_n - \boldsymbol{W}\boldsymbol{s}_n)\right\} \times \\
&\qquad \sum_{k=1}^{K} \pi_k |2\pi\boldsymbol{\Sigma}_k|^{-\frac{1}{2}} \exp\left\{-\frac{1}{2}(\boldsymbol{s}_n - \boldsymbol{\mu}_k)^T\boldsymbol{\Sigma}_k^{-1}(\boldsymbol{s}_n - \boldsymbol{\mu}_k)\right\}.
\end{aligned} \tag{30}
$$

By calculating the expectation values

$$\gamma_{nk} \equiv p(k|\boldsymbol{x}_n) \tag{31}$$

$$\langle \boldsymbol{s}_n \rangle_k = \int p(\boldsymbol{s}|\boldsymbol{x}_n, k)\, \boldsymbol{s}\, |d\mathbf{s}| \tag{32}$$

$$\langle \boldsymbol{s}_n \rangle = \int p(\boldsymbol{s}|\boldsymbol{x}_n)\, \mathbf{s}\, |d\boldsymbol{s}| \tag{33}$$

$$\langle(\boldsymbol{s}_n - \langle\boldsymbol{s}_n\rangle)(\boldsymbol{s}_n - \langle\boldsymbol{s}_n\rangle)^T\rangle = \sum_k p(k|\boldsymbol{x}_n)\,\mathbf{M}_k, \tag{34}$$

we can obtain the expectation of the complete data log-likelihood in the form of

$$
\begin{aligned}
\langle \mathcal{L}_C \rangle \;=\; & \sum_{n=1}^{N}\sum_{k=1}^{K} \gamma_{nk} \{ \ln \pi_k - \frac{1}{2}\ln|\mathbf{\Sigma}_k| - \frac{1}{2}\mathrm{tr}(\mathbf{\Sigma}_k^{-1}\langle \boldsymbol{s}_n \boldsymbol{s}_n^T \rangle) + \boldsymbol{\mu}_k^T \mathbf{\Sigma}_k^{-1}\langle \boldsymbol{s}_n \rangle - \frac{1}{2}\boldsymbol{\mu}_k^T \mathbf{\Sigma}_k^{-1}\boldsymbol{\mu}_k \\
& - \frac{1}{2}\ln|\mathbf{\Psi}| - \frac{1}{2}\boldsymbol{x}_n^T \mathbf{\Psi}^{-1}\boldsymbol{x}_n + \boldsymbol{x}_n^T \mathbf{\Psi}^{-1}\boldsymbol{W}\langle \boldsymbol{s}_n \rangle - \frac{1}{2}\mathrm{tr}(\boldsymbol{W}^T \mathbf{\Psi}^{-1}\boldsymbol{W}\langle \boldsymbol{s}_n \boldsymbol{s}_n^T \rangle) \} \\
& - \lambda(\sum_{k=1}^{K}\pi_k - 1)
\end{aligned}
\tag{35}
$$

where the expectation values are given by

E-steps:

$$
\gamma_{nk} \;=\; \frac{\pi_k \mathcal{N}(\boldsymbol{x}_n | \boldsymbol{W}\boldsymbol{\mu}_k, \boldsymbol{V}_k)}{\sum_{k'} \pi_{k'} \mathcal{N}(\boldsymbol{x}_n | \boldsymbol{W}\boldsymbol{\mu}_{k'}, \boldsymbol{V}_{k'})}
\tag{36}
$$

$$
\langle \boldsymbol{s}_n \rangle_k \;=\; \boldsymbol{\mu}_k + \boldsymbol{M}_k \boldsymbol{W}^T \mathbf{\Psi}^{-1}(\boldsymbol{x}_n - \boldsymbol{W}\boldsymbol{\mu}_k)
\tag{37}
$$

$$
\langle \boldsymbol{s}_n \rangle \;=\; \sum_k \gamma_{nk}\langle \boldsymbol{s}_n \rangle_k
\tag{38}
$$

$$
\langle \boldsymbol{s}_n \boldsymbol{s}_n^T \rangle \;=\; \sum_k \gamma_{nk}\left( \boldsymbol{M}_k + (\langle \boldsymbol{s}_n \rangle_k - \langle \boldsymbol{s}_n \rangle)(\langle \boldsymbol{s}_n \rangle_k - \langle \boldsymbol{s}_n \rangle)^T \right) + \langle \boldsymbol{s}_n \rangle \langle \boldsymbol{s}_n \rangle^T
\tag{39}
$$

with $\boldsymbol{V}_k = \mathbf{\Psi} + \boldsymbol{W}\mathbf{\Sigma}_k \boldsymbol{W}^T$ and $\boldsymbol{M}_k = (\mathbf{\Sigma}_k^{-1} + \boldsymbol{W}^T \mathbf{\Psi}^{-1}\boldsymbol{W})^{-1}$.

Equation (35) is maximized by the following M-step formulas

M-steps:

$$
\pi_k \;=\; \frac{\sum_n \gamma_{nk}}{N}
\tag{40}
$$

$$
\boldsymbol{\mu}_k \;=\; \frac{1}{\sum_n \gamma_{nk}}\sum_n \gamma_{nk}\langle \boldsymbol{s}_n \rangle
\tag{41}
$$

$$
\mathbf{\Sigma}_k \;=\; \frac{1}{\sum_n \gamma_{nk}}\sum_n \gamma_{nk}(\langle \boldsymbol{s}_n \boldsymbol{s}_n^T \rangle - \boldsymbol{\mu}_k \langle \boldsymbol{s}_n \rangle^T - \langle \boldsymbol{s}_n \rangle \boldsymbol{\mu}_k^T + \boldsymbol{\mu}_k \boldsymbol{\mu}_k^T)
\tag{42}
$$

$$
\boldsymbol{W} \;=\; (\sum_n \boldsymbol{x}_n \langle \boldsymbol{s}_n \rangle^T)(\sum_n \langle \boldsymbol{s}_n \boldsymbol{s}_n^T \rangle)^{-1}
\tag{43}
$$

$$
\mathbf{\Psi} \;=\; \mathrm{Diag}(\sum_{n=1} \boldsymbol{x}_n \boldsymbol{x}_n^T - 2\boldsymbol{W}\langle \boldsymbol{s}_n \rangle \boldsymbol{x}_n^T + \boldsymbol{W}\langle \boldsymbol{s}_n \boldsymbol{s}_n^T \rangle \boldsymbol{W}^T)N^{-1}
\tag{44}
$$

where $\mathrm{Diag}(\cdot)$ returns a diagonal matrix composed of the diagonal elements of a given square matrix.

---

**Algorithm 1** Gaussian Mixture Counterfactual Generator (GMCG)

---

**Require:** (1) Factual training data set $\{\boldsymbol{x}_{n,\mathrm{f}}\}$; (2) Factual test data $\boldsymbol{x}_\mathrm{f}$ and alternative treatment $\boldsymbol{a}_\mathrm{cf}$
**Ensure:** (1) GMCG parameters $\mathbb{Q} = \{\{\pi_k\}, \{\boldsymbol{\mu}_k\}, \{\boldsymbol{\Sigma}_k\}, \boldsymbol{W}, \boldsymbol{\Psi}\}$; (2) Counterfactual data $\boldsymbol{x}_\mathrm{cf}$

---

 1: **Training an SSA model**
 2: Input: $\{\boldsymbol{x}_{n,\mathrm{f}}\}$
 3: Initialize $\mathbb{Q}$
 4: **repeat**
 5:      # E-step
 6:      **for** $n = 1$ to $N$ **do**
 7:          **for** $k = 1$ to $K$ **do**
 8:              Compute $\gamma_{nk}$ and $\langle \boldsymbol{s}_n \rangle_k$ from Eqs. (36) and (37)
 9:          **end for**
10:          Compute $\langle \boldsymbol{s}_n \rangle$ and $\langle \boldsymbol{s}_n \boldsymbol{s}_n^T \rangle$ from Eqs. (38) and (39)
11:      **end for**
12:      # M-step
13:      **for** $k = 1$ to $K$ **do**
14:          Compute $\pi_k$, $\boldsymbol{\mu}_k$, and $\boldsymbol{\Sigma}_k$ from Eqs. (40), (41), and (42)
15:      **end for**
16:      Compute $\boldsymbol{W}$ and $\boldsymbol{\Psi}$ from Eqs. (43) and (44)
17: **until** convergence
18: **return** $\mathbb{Q}$

19: **Inferring a new patient's state vector**
20: Input: $\mathbb{Q}$, $\boldsymbol{x}_\mathrm{f}$
21: **for** $k = 1$ to $K$ **do**
22:      Compute $\gamma_k$ and $\langle \boldsymbol{s} \rangle_k$ from Eqs. (36) and (37)
23: **end for**
24: Compute $\langle \boldsymbol{s} \rangle$ from Eq. (38)
25: **return** $\langle \boldsymbol{s} \rangle$ as $\boldsymbol{s}_\mathrm{f}$

26: **Generating the patient's counterfactual data**
27: Input: $\mathbb{Q}$, $\boldsymbol{x}_\mathrm{f}$, $\boldsymbol{s}_\mathrm{f}$, $\boldsymbol{a}_\mathrm{cf}$
28: Compute $\boldsymbol{\delta}_a$ and $\boldsymbol{N}$ from Eq. (14)
29: **for** $k = 1$ to $K$ **do**
30:      Compute $\boldsymbol{C}_k$, $\boldsymbol{m}_k$, and $p_k$ from Eqs. (16), (17), and (18)
31: **end for**
32: Generate $\boldsymbol{x}_\mathrm{cf}$ from Eq. (19)
33: **return** $\boldsymbol{x}_\mathrm{cf}$

---

## C  ADDITIONAL MATERIALS FOR CROSSOVER TRIAL EXAMPLE

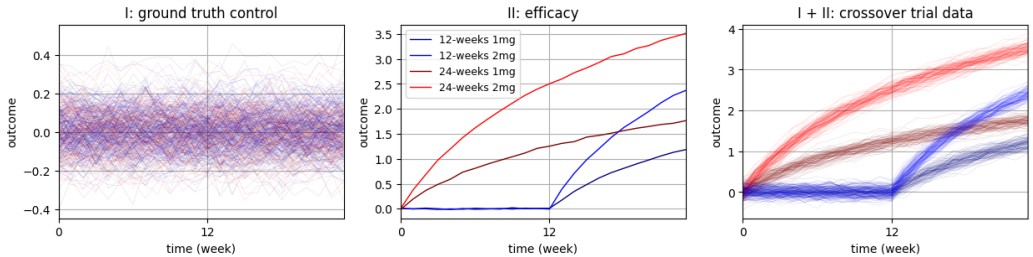

Figure 2: A simple illustrative example of creating crossover trials data.

**Data Generation**.  The linear efficacy data presented in Figure 1 were created using the method shown in Figure 2. First, as illustrated in the left panel, outcome data for 400 patients were created by adding noise mainly within the range of -0.2 to 0.2. Figure 1 used these data as the ground-truth control. The middle panel shows the assumed efficacy for four groups based on the duration and dosage of the drug used. It was assumed that all patients in each group exhibited the same efficacy. The assumed curve shape was generated as a function of time, specifically using $\log(t + 1)$ for $0 \leq t < 24$. The two blue-shaded curves are horizontal translations of the two red-shaded curves. The final synthetic crossover trial data, shown in the right panel, were created by combining the data from panels I and II for each patient. This way represents the simplest form of data and does not accurately reflect accurate clinical trial data. Clinical data include discontinuous values, missing values, and various complex covariate structures. While covariate data can provide additional helpful information for experimental results, they were excluded from this example.

**Nonlinear Efficacy**.  In Figure 1, the synthetic control data arm could be created with only two different dose groups per arm due to the assumption that the efficacy is proportional to the dose administered. However, in a more general case, where the relationship between dose and efficacy is nonlinear, the generation of synthetic control data requires more dose groups. Figure 3 presents an example that addresses this question, showing four dose groups for each arm: shades of red representing four dose groups with 0.5, 1.0, 1.5, and 2.0 mg over the full 24-week period, and shades of blue representing four other groups with the same doses during only the last 12 weeks.

The nonlinear relationship assumed here is modeled by $(\text{treatment effect by a dose}) = (\text{dose} + 0.2 \times \text{dose}^2) \times (\text{treatment effect by 1mg dose})$. Each group consists of 100 individuals, totaling 800 training data. The four small plots on the right show synthetic control data for four hypothetical patients taking 1.0 mg and 2.0 mg in each arm, with the generated synthetic mean control data closely approximating the ground truth control.

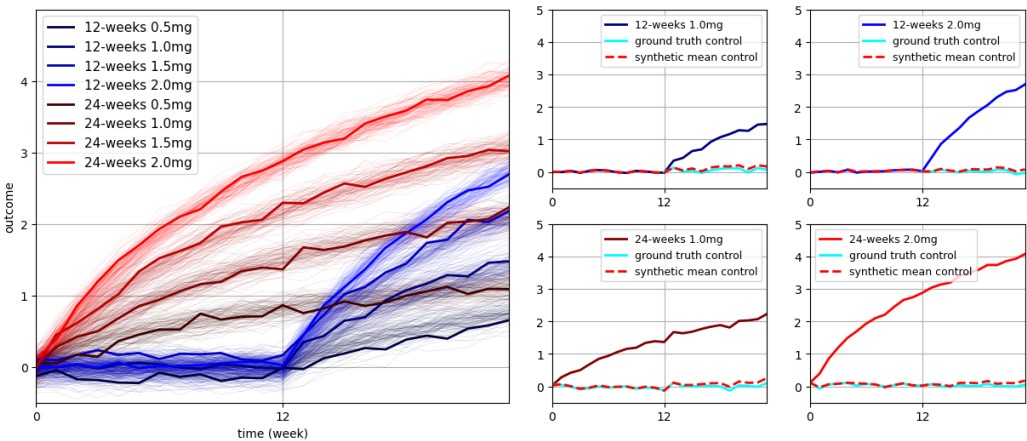

Figure 3: Simulated crossover trial dataset (nonlinear efficacy) and synthetic control data generation.

This problem's nonlinear effect cannot be adequately captured with only two dose groups per arm. The Appendix provides additional results related to this issue. This situation is similar to needing at least three points, rather than just two, to determine a quadratic function's position, direction, and curvature. Therefore, more diverse dose levels are necessary in the experimental data to uncover complex or hidden effects in counterfactual predictions. Therefore, more diverse dose levels are likely necessary in the experimental data to uncover more complex or hidden effects in counterfactual predictions.

This section demonstrates how the proposed GMCG algorithm can be applied to crossover trial data and the types of results it can produce. However, it is important to note that this paper does not claim guaranteed success for all types of crossover trials. As an algorithm within the GMM family, the GMCG algorithm's clear and concise mathematical model highlights its potential to tackle new types of synthetic data generation problems. Discussing this potential alone represents a meaningful achievement.

**Non-trivial Control** The ground-truth control data in Figure 1 or 2 follows a simple linear trend, which may seem trivial. However, the proposed method does not fit the shape or distribution of the ground-truth control. Figure 3 demonstrates that the ground-truth control data is well predicted even when following a sine wave pattern.

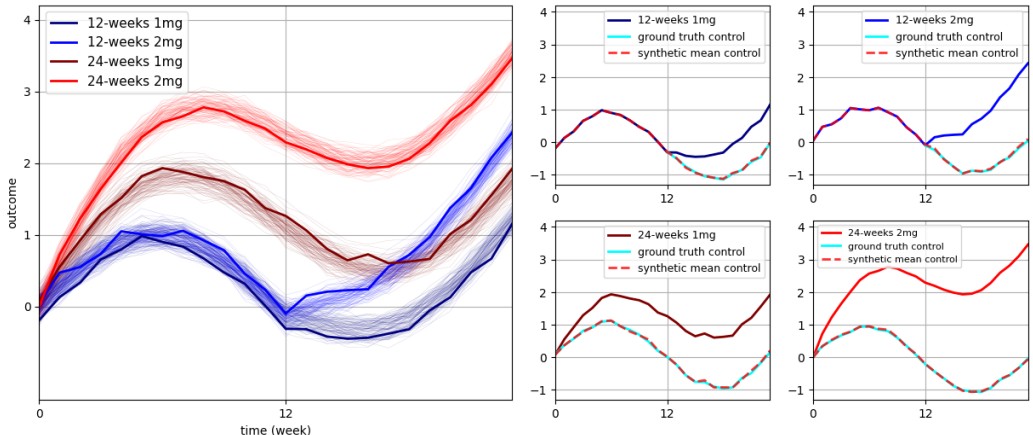

Figure 4: Simulated crossover trial dataset (linear efficacy and sinusodial ground truth control) and synthetic control data generation.

The problem of estimating the nonlinear effect of dosage shown in Figure 3 cannot be adequately addressed with only two dose groups per arm. Therefore, Figure 3 uses four dose groups per arm. Figure 5 shows the bad result when only two dose groups are used. It demonstrates that the synthetic control for the 12-week treatment group is pulled upward due to the nonlinear effect, and the 24-week treatment group shows unstable predictions with significant fluctuations. Increasing the diversity of dosages in the training data, i.e., using more dose groups per arm, can achieve more stable and accurate counterfactual predictions. For instance, if a hypothetical experiment is conducted where each patient receives slightly different dosages, as shown in Figure 6, the results improve even in more complex nonlinearities in efficacy.

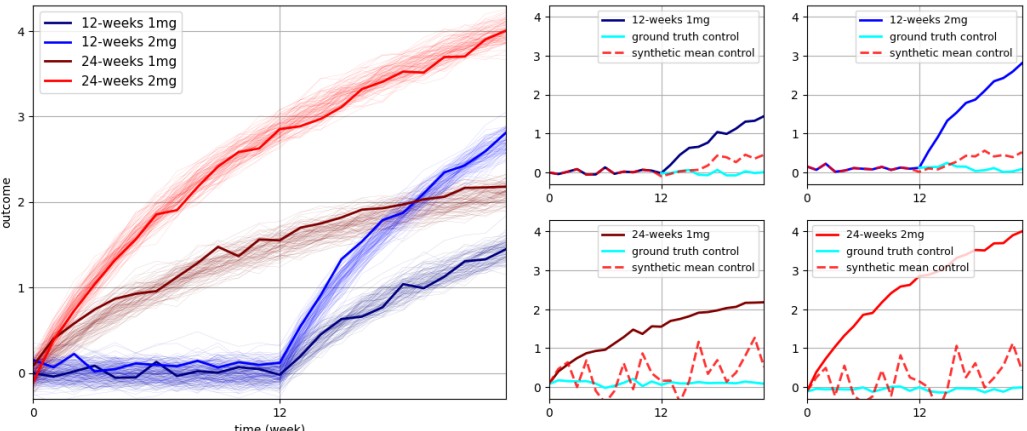

Figure 5: Simulated crossover trial dataset (nonlinear efficacy) and incorrect synthetic control data generation.

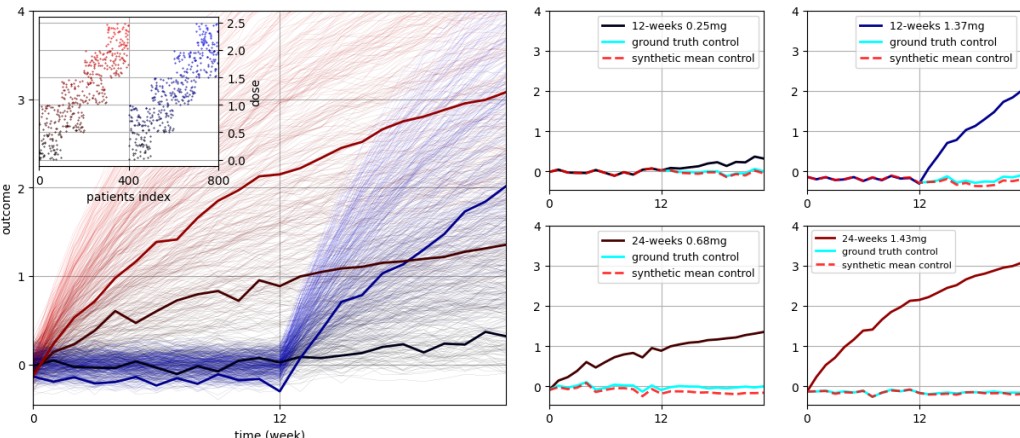

Figure 6: Simulated crossover trial dataset (nonlinear efficacy with diverse doses) and synthetic control data generation.

# D  ADDITIONAL RESULTS OF LDL DATA EXPERIMENTS

In this experiment, we did not use dimensionality reduction. Thus, $W$ becomes an identity matrix, making the EM algorithm of section B equivalent to the standard GMM algorithm. Specifically, we used one component for each of the treatment and control groups, which trivially means performing single Gaussian density estimation for each group. At this point, there is still an undetermined noise model: $\Psi = \sigma^2 I$ (see Appendix B to check how $\Psi$ affects the parameters of Eq. (19)), and we manually substituted various $\sigma$ values to obtain the counterfactual predictions given by Eq. (19). The results are shown in Fig. 7. The figure shows that very small or very large sigma values resulted in high error values, while values between 0.1 and 1 yielded the lowest error values. Considering that we added noise with a variance of 0.1 when simulating LDL data, it explains why the lowest error value in terms of MAE was obtained around 0.1.

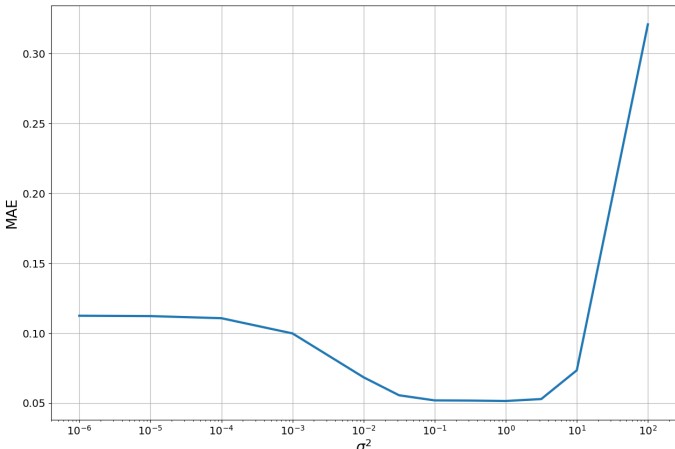

Figure 7: Counterfactual prediction error (MAE) as a function of $\sigma^2$ in the noise covariance model $\Psi = \sigma^2 I$.

Figure 8 plots 15 different patients' examples obtained by applying the GMCG algorithm to the LDL cholesterol data. Specifically, for a total time length of $T = 30$, the simulation was performed at $t = 25$ when statin medication was taken, and the factual data, ground truth counterfactual data, and counterfactual prediction obtained through the GMCG algorithm were plotted on each axis. There are two points of observation here: the pretreatment data before $t = 25$ align exactly with the factual data, and the post-treatment data after $t = 25$ closely approximate the ground-truth counterfactual data.

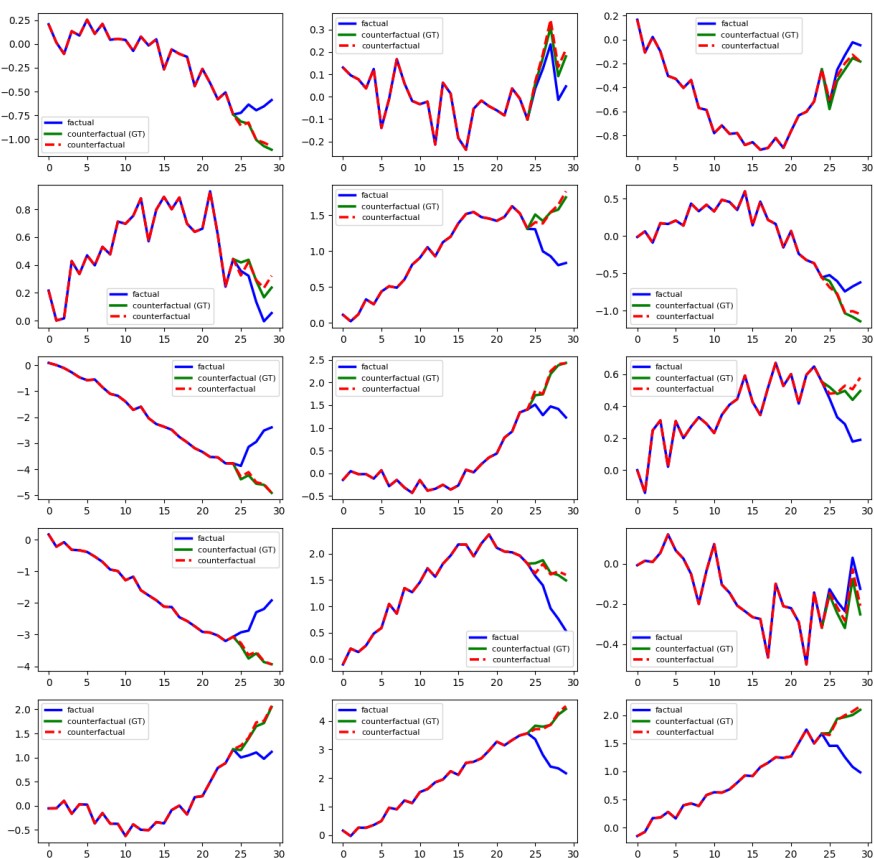

Figure 8: Several plots derived from the results of $p_0 = 0.1$ and $N = 400$ presented in Table 1. Each subplot represents the data of an individual patient. There are two key points to note here: First, the pre-treatment data before $t = 25$ matches exactly between the factual data and counterfactual prediction. Second, the counterfactual prediction after $t = 25$ closely approximates the given ground truth post-treatment data. This serves as further evidence of the excellent results shown in Table 1.

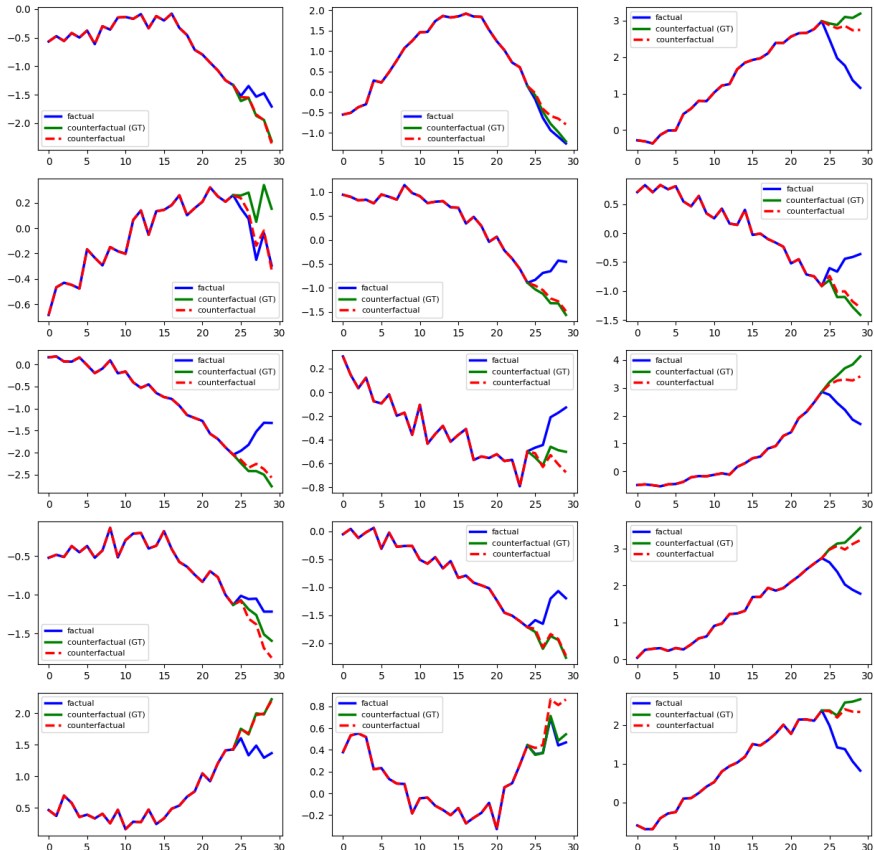

Figure 9: Several plots derived from the results of $p_0 = 0.5$ and $N = 1200$ presented in Table 2. Each subplot represents the data of an individual patient. There are three key points to note here: First, these results were generated by a GMCG model trained without control data, generating control data. Second, the pre-treatment data before $t = 25$ matches exactly between the factual data and counterfactual prediction. Third, the counterfactual prediction after $t = 25$ approximates the given ground truth post-treatment data. This serves as further evidence of the excellent results shown in Table 1.

# E  SIMULATED TUMOR GROWTH DATA

Although the proposed algorithm is designed to generate synthetic control arms, it can also perform general causal inference, as defined in Section 3.3. This definition aligns with the problem definitions specified in the RMSN and CRN algorithms (Lim, 2018; Bica et al., 2020b). Therefore, in this section, we applied the algorithm to tumor growth data (Geng et al., 2017), which is commonly used for validating general causal inference algorithms. We used data from approximately 10,000 virtual patients and applied it to unbiased and biased data for $T = 10, 20, 30$. The number of mixture components used in the GMCG model is $K = 30$. We compared the state-of-the-art CRN model to the same data and obtained the results. Detailed information is provided in the Appendix.

Table 3 shows the mean absolute error (MAE) obtained by varying $\delta$ while keeping $\tau$ fixed for data with $T = 10$ and $T = 20$. The error measure was averaged between $t = \tau + 1$ and $t = \tau + \delta$. Surprisingly, the GMM-based GMCG algorithm outperformed the RNN-based CRN regarding MAE. Results for unbiased data are also summarized in the Appendix.

| $(T, \tau)$ | $(10, 5)$ | | | | | $(20, 15)$ | | | | |
|---|---|---|---|---|---|---|---|---|---|---|
| $\delta$ | 1 | 2 | 3 | 4 | 5 | 1 | 2 | 3 | 4 | 5 |
| GMCG | 0.0436 | 0.0484 | 0.0520 | 0.0505 | 0.0497 | 0.0044 | 0.0051 | 0.0054 | 0.0054 | 0.0056 |
| CRN | 0.0587 | 0.0596 | 0.0627 | 0.0633 | 0.0644 | 0.0325 | 0.0283 | 0.0283 | 0.0292 | 0.0305 |

Table 3: Mean absolute errors (MAE) comparison for biased tumor growth data.

As the duration $T$ increases, there is a tendency for the error values to decrease. However, this cannot be solely attributed to the more extended data being learned. Fundamentally, with longer durations, the tumor is exposed to more therapy sessions, and the tumor size has already reduced. Consequently, the ground truth counterfactual tumor size for other arbitrary therapy alternatives does not differ significantly from factual tumor size. While the GMCG results closely approximated the ground truth counterfactual data or factual data, the CRN's counterfactual predictions showed some differences from both of them.

How can a GMM-based algorithm yield better results than an RNN-based adversarial network algorithm? Firstly, it is important to recall that GMM can act as an excellent universal approximator as the number of mixture components increases (Heaton, 2018). Additionally, Problem I defined in Section 3.2 is an unsupervised learning problem, and GMM is stably optimized through the EM algorithm (McLachlan & Krishnan, 2007). The most crucial factor lies in the *conditional* probability calculated through Equation (15). Even in regions with sparse training data, the conditional probability calculates a *normalized* probabilistic density distribution conditional on the counterfactual assumption, which creates a robust characteristic against bias. This explains the successful generation of a synthetic control arm in the crossover trials example where training data did not exist.

# F   ADDITIONAL MATERIALS FOR TUMOR DATA EXAMPLE

## F.1   USED HYPERPARAMETERS

To train the CRN for our experiments, we used the default hyperparameters. The exact values used are as follows. These values fall within the optimized hyper-parameters range in Tables 6 and 7 of Appendix J in the CRN paper (Bica et al., 2020b).

The encoder's used hyperparameters:

$$\text{num\_epochs} = 100$$
$$\{ \text{rnn\_hidden\_units} : 24,$$
$$\text{br\_size} : 12,$$
$$\text{fc\_hidden\_units} : 36,$$
$$\text{learning\_rate} : 0.01,$$
$$\text{batch\_size} : 128,$$
$$\text{rnn\_keep\_prob} : 0.9 \}$$

The decoder's used hyperparameters:

$$\text{num\_epochs} = 100$$
$$\{ \text{rnn\_hidden\_units} : 12,$$
$$\text{br\_size} : 18,$$
$$\text{fc\_hidden\_units} : 36,$$
$$\text{learning\_rate} : 0.001,$$
$$\text{batch\_size} : 1024,$$
$$\text{rnn\_keep\_prob} : 0.9 \}$$

where (br_size) is (balancing representation size) and (rnn_keep_prob) = 1 - (RNN dropout probability).

## F.2   ADDITIONAL TABLE

| $(T, \tau)$ | $(10, 5)$ | | | | | $(20, 15)$ | | | | |
|---|---|---|---|---|---|---|---|---|---|---|
| $\delta$ | 1 | 2 | 3 | 4 | 5 | 1 | 2 | 3 | 4 | 5 |
| GMCG | 0.0548 | 0.0636 | 0.0697 | 0.0694 | 0.0682 | 0.0057 | 0.0061 | 0.0071 | 0.0071 | 0.0073 |
| CRN | 0.0831 | 0.0795 | 0.0773 | 0.0788 | 0.0776 | 0.0429 | 0.0390 | 0.0348 | 0.0327 | 0.0296 |

Table 4: Mean absolute errors (MAE) comparison for unbiased tumor growth data for $T = 10$ and $T = 20$.

| $(T, \tau)$ | $(30, 15)$ | | | | | $(30, 25)$ | | | | |
|---|---|---|---|---|---|---|---|---|---|---|
| $\delta$ | 1 | 2 | 3 | 4 | 5 | 1 | 2 | 3 | 4 | 5 |
| GMCG | 0.0075 | 0.0072 | 0.0077 | 0.0079 | 0.0081 | 0.0015 | 0.0017 | 0.0019 | 0.0019 | 0.0020 |
| CRN | 0.0179 | 0.0226 | 0.0262 | 0.0293 | 0.0321 | 0.0128 | 0.0151 | 0.0167 | 0.0192 | 0.0209 |

Table 5: Mean absolute errors (MAE) comparison for unbiased tumor growth data for $T = 30$.

# G   ASSUMPTIONS

## G.1   STANDARD ASSUMPTIONS

This paper adopts Rubin's potential outcome framework for LDL cholesterol data analysis presented in Table 1 and is based on standard assumptions commonly found in many studies, such as consistency, overlap, and unconfoundedness. In addition, important new assumptions are discussed in Section 4.1, where we explore the concept of counterfactual reasoning. There are two types of counterfactual reasoning: retrospective counterfactual reasoning, which examines the effects of altering past treatments at the current time, and prospective counterfactual reasoning, which forecasts the outcomes of choosing among various possible future treatments at the present moment.

**Assumption 1: Consistency**. If treatment $\mathbf{a}^{(:t)} = \boldsymbol{a}_{\mathrm{f}}^{(:t)}$ was administered to a given patient, then the potential outcome for treatment $\boldsymbol{a}_{\mathrm{f}}^{(:t)}$ is the same as the observed factual outcome: $\boldsymbol{y}_{\mathrm{f}}^{(t+1)} = \boldsymbol{y}^{(t+1)}[\boldsymbol{a}_{\mathrm{f}}^{(:t)}]$.

**Assumption 2: Positivity**. If $p(\mathbf{x}^{(:t)} = \boldsymbol{x}^{(:t)}) \neq 0$, then $p(\mathbf{a}^{(t)} = \boldsymbol{a}^{(t)}|\mathbf{x}^{(:t)} = \boldsymbol{x}^{(:t)}) > 0$ for all $\boldsymbol{a}^{(t)}$.

**Assumption 3: Sequential Strong Ignorability**. $\mathbf{y}^{(t+1)}[\boldsymbol{a}^{(t)}] \perp\!\!\!\perp \mathbf{a}^{(t)}|\mathbf{x}^{(:t)}$ for all $\boldsymbol{a}^{(t)}$.

## G.2   ALTERNATIVE ASSUMPTIONS FOR POSITIVITY VIOLATION

When the positivity assumption is violated, as with the data used in Table 2, an alternative assumption is needed. Generally, it can be assumed that the conditional expectation of the outcome variable is a sufficiently smooth function with respect to the treatment variable. This continuity allows for the estimation of outcomes for treatment values that are not directly observed, i.e.,

**Assumption 4: Strong Continuity**. $\mathbb{E}[\mathbf{y}|\mathbf{x}_{\mathrm{f}} = \boldsymbol{x}_{\mathrm{f}}, \mathbf{a}_{\mathrm{cf}} = \boldsymbol{a}_{\mathrm{cf}}]$ is continuous and differentiable with respect to treatment to an appropriate order.

Alternatively, it can be assumed that the relationship between the outcome $\mathbf{y}$, the counterfactual treatment $\mathbf{a}_{\mathrm{cf}}$, and the covariates $\mathbf{x}_{\mathrm{f}}$ follows a specific functional form (e.g., linear or quadratic regression model). This assumption enables extrapolation to estimate causal effects:

**Assumption 5: Functional Form of Outcome Model**. $\mathbb{E}[\mathbf{y}|\mathbf{x}_{\mathrm{f}} = \boldsymbol{x}_{\mathrm{f}}, \mathbf{a}_{\mathrm{cf}} = \boldsymbol{a}_{\mathrm{cf}}] = f(\boldsymbol{x}_{\mathrm{f}}, \boldsymbol{a}_{\mathrm{cf}}; \boldsymbol{\theta})$ where $f$ is a known functional form, and $\boldsymbol{\theta}$ represents parameters to be estimated.

Specifically, the results in Figure 1 and the various examples in the Appendix were obtained based on this assumption.

## H    PROOF OF GAUSSIAN MIXTURE COUNTERFACTUAL GENERATOR

**Lemma 1.** *Let the random vector $\xi$ follow a mixture of multivariate normal distributions with the $k$-th mean $\boldsymbol{m}_k$ and the $k$-th covariance matrix $\boldsymbol{C}_k$ for $k = 1, \cdots, K$. Let $\boldsymbol{A}$ be a full-rank matrix and $\boldsymbol{b}$ be a translation vector. Then the random vector $\mathbf{x}$ defined by*

$$\mathbf{x} = \boldsymbol{A}\xi + \boldsymbol{b}$$

*has a mixture of multivariate normal distributions with the $k$-th mean*

$$\boldsymbol{A}\boldsymbol{m}_k + \boldsymbol{b} \tag{45}$$

*and the $k$-th covariance matrix*

$$\boldsymbol{A}\boldsymbol{C}_k\boldsymbol{A}^T \tag{46}$$

*with the same proportions.*

*Proof.* See Johnson et al. (2002) for the proof.    □

### H.1    PROOF OF GMCG

*Proof.* Subtracting Eq. (6) from Eq. (9), we obtain the following equation:

$$\mathbf{x}_{\text{cf}}^{(\tau:)} - \boldsymbol{x}_{\text{f}}^{(\tau:)} = \boldsymbol{W}^{(\tau:)}(\mathbf{s}_{\text{cf}} - \boldsymbol{s}_{\text{f}}).$$

Combining this with Eq. (13), we can write it as follows:

$$\mathbf{x}_{\text{cf}} = \boldsymbol{x}_{\text{f}} + \boldsymbol{W}(\mathbf{s}_{\text{cf}} - \boldsymbol{s}_{\text{f}}).$$

Using Eq. (14) to simplify the expression and rearranging the equation with respect to $\xi$, we can organize the equation in the form of $\boldsymbol{b} + \boldsymbol{A}\xi$:

$$\begin{aligned}
\mathbf{x}_{\text{cf}} &= \boldsymbol{x}_{\text{f}} + \boldsymbol{W}(\boldsymbol{\delta}_a + \boldsymbol{N}\xi) \\
&= (\boldsymbol{x}_{\text{f}} + \boldsymbol{W}\boldsymbol{\delta}_a) + (\boldsymbol{W}\boldsymbol{N})\xi \\
&\equiv \boldsymbol{b} + \boldsymbol{A}\xi.
\end{aligned}$$

Using Eq. (45) and (46) of Lemma 1, we finally obtain Eq. (19):

$$\mathbf{x}_{\text{cf}} \sim \sum_{k=1}^{K} p_k \mathcal{N}\left(\mathbf{x}; \boldsymbol{x}_{\text{f}} + \boldsymbol{W}(\boldsymbol{\delta}_a + \boldsymbol{N}\boldsymbol{m}_k), \boldsymbol{W}\boldsymbol{N}\boldsymbol{C}_k\boldsymbol{N}^T\boldsymbol{W}^T\right).$$

□

# I    LIMITATION OF SCM IN CREATING A SYNTHETIC CONTROL ARM

Generating synthetic control arms in crossover trials is challenging due to the unavailability of control group data after patients switch to active treatment. In a standard RCT, control group data are crucial for comparing outcomes over time. However, in crossover trials—especially those with an open-label extension phase—patients initially assigned to the control group often switch to the active treatment after a certain period (e.g., after 12 weeks). This switch means that there are no remaining patients in the control group during the extended phase (e.g., up to 24 weeks). The absence of extended control data makes it difficult to assess the long-term efficacy and safety of the treatment, as there is no direct comparison group.

Abadie's synthetic control method (SCM) attempts to approximate the treatment group's outcomes by creating a weighted combination of control units that closely resembles the treatment-assigned patient data before the intervention (Abadie & Gardeazabal, 2003; Abadie et al., 2010). This method relies on the availability of control data to compute the necessary weights. Applying Abadie's method becomes problematic in the context of crossover trials lacking extended control data. For example, in a trial with a 12-week RCT followed by a 12-week open-label extension where all patients receive the active treatment, no 24-week control data is available. Without this data, we cannot calculate the weighted sum of the 24-week control outcomes needed to construct a synthetic control arm for the switched patients. This limitation prevents the direct application of Abadie's algorithm in such scenarios. In Zhou et al. (2024), SCM and difference-in-differences (Abadie, 2005; Sant'Anna & Zhao, 2020) methods successfully measured treatment effects in crossover trials with the availability of external controls; however, this approach is not feasible in studies where securing external controls is challenging.

