# OpenReview forum: "Gaussian Mixture Counterfactual Generator"
_ICLR.cc/2025/Conference — ICLR 2025 Poster_

### Official Review · Reviewer_zi3C · 2024-10-31

**Soundness:** 2
**Presentation:** 2
**Contribution:** 2
**Rating:** 8
**Confidence:** 3

**Summary:**

Problem: counterfactual data may not exist in the extended phase of a crossover trial, so synthetic control methods cannot be used to assess treatment efficacy with respect to a no-treatment baseline.
Significance: makes it difficult to assess long-term efficacy and safety of treatments in this setting.
Solution: the authors propose a model (the Gaussian Mixture Counterfactual Generator) to generate the synthetic controls.

**Strengths:**

- a, to my knowledge, novel approach to a real problem.
- technically sound , though perhaps missing important assumptions
- illustrative experiments

**Weaknesses:**

The main weakness of this paper is the lack of formal assumptions explaining when this method will yield accurate results. From the results presented it seems like the fidelity of the generated controls will depend on how well the model estimates the dose response function, and how well it can extrapolate to the 0 dose condition. At least this seems sensible to me after reviewing the linear relationship in Figure 1 and the non-linear relationship analysis in Figures 2, 5, and 6.

The empirical results could be strengthened by including simulated but realistic placebo models. Perhaps adding toxicity models for dose responses.

**Questions:**

Can you formalize a condition/assumption that would cover the failure mode illustrated in Figure 5? I think this will be a necessary addition for me to consider this paper for acceptance, and I think it is doable.

Are there realistic placebo models that you could incorporate into your synthetic experiments?

Is your non-linear dose response function a realistic one? Are there other functions like toxicity models that you could add?

In the absence of a formal condition on the dose response and perhaps placebo or 0 dose models, i would also accept a more thorough empirical evaluation. If the answers to the above questions are affirmative, then those experiments could be added, along with varying the number of patients. This would allow practitioners to have a fuller picture of when this approach might be appropriate for their problem.

---

> ### Author Response · Authors · 2024-11-25
>
> Thank you for your insightful comments. Following your suggestions, we conducted new experiments to improve our paper and included the results of our quantitative analysis. Please review the content and tables in Section 5.2.
>
> The data was simulated using a PK/PD model related to statins and their effects on LDL cholesterol levels. We carried out experiments with patient samples of $N$=400, 1200, and 2000, and we also varied the confounding bias with $p_0$ = 0.1, 0.25, and 0.5. The results of our comparative experiment to the SSA-GMM model, which was the starting point for this paper, are included in Table 1, and our results outperform all the other algorithms.
>
> Table 2 shows the results of crossover trials conducted without control data, comparing them with the results in Table 1. These results are encouraging. As you pointed out, this is a valuable simulation that enables practitioners to understand the algorithm and its characteristics. If you find the mathematical prototype of the GMCG algorithm, the simulated crossover trials utilizing LDL cholesterol data, and the results to be valid, we kindly ask you to consider a higher rating.
>
> - Question
>
> [1] "Can you formalize a condition/assumption that would cover the failure mode illustrated in Figure 5? I think this will be a necessary addition for me to consider this paper for acceptance, and I think it is doable."
>
> => The example in Figure 5 is based on data created assuming a quadratic relationship between dose and outcome. Since determining a quadratic function requires three coefficients, generally, three conditions are needed to determine it uniquely.
>
> However, the data in Figure 5 has a minimum dose representation dimension of 2D, not 1D, because it is divided into two groups: 12 weeks and 24 weeks. In other words, the raw representation of the dose is 24-dimensional. But since the same dose is applied to patients for either 12 or 24 weeks, there is redundancy, and the dimension can be compressed to 2D.
>
> Therefore, there are two quadratic relationships, meaning a total of six coefficients need to be determined. Thus, there must be at least six subgroups. However, Figure 5 has only four subgroups, making it an underdetermined problem, resulting in unstable values. Generally, the more subgroups receiving the same dose, the better. Hence, in Figure 6, we conducted a hypothetical experiment with continuously given dose levels, and the results were satisfactory. Similarly, since statins have a highly nonlinear dose-response, we also subdivided the dose levels continuously in the newly added LDL cholesterol data experiment in section 5.2.2.
>
> Additionally, all patients should not show the same outcome pattern for the same dose. If the data clusters into a single point in space, the covariance matrices of the GMM model become singular. Even if the dose levels are diversified, if the patient response lacks diversity, the covariance matrices of the GMM will not have a good condition number. Diversity in patient responses is also crucial. However, this is not an issue with real-world data.

---

> > ### Comment · Reviewer_zi3C · 2024-11-26
> >
> > Thank you for addressing my concerns. I am happy to change my score to accept.

---

> > > ### Author Response · Authors · 2024-11-28
> > >
> > > Thank you for evaluating our paper positively. We are very pleased that your thorough review and the identification of the shortcomings in the experimental section have allowed us to present more convincing experimental results.

---

### Official Review · Reviewer_jKvu · 2024-11-01

**Soundness:** 4
**Presentation:** 3
**Contribution:** 3
**Rating:** 8
**Confidence:** 4

**Summary:**

This paper considers causal inference for cross-over trials (and some other types of settings). Cross-over trials are settings in which at some fixed point partway through the trial, the placebo group recieve treatment. Causal inference in these settings is complicated by the fact that in later periods there is no longer a placebo group with which to compare treated outcomes. This renders counterfactual prediction methods like synthetic control (Abadie and Garbanzal 2003) inapplicable. In order to simulate counterfactual no-treatment outcomes in later periods, the authors suggest the use of novel method based on Gaussian mixture models. For each individual, the path for all variables, including the treatment and outcome, are drawn from a Gaussian model that depends on an individual-specific and time-invariant state-vector s. In order to simulate counterfactuals, the researchers propose (implicitly) drawing s from its conditional distribution under a counterfactual path for the treatment variable that is not seen in the data and drawing other variables conditional on this s and the counterfactual treatment path.

 As I understand it, the method is essentially an imputation approach. We do not observe individuals who are never treated in the data, but by imposing a sufficiently restrictive model for the relationship between all variables including treatment, we can impute outcomes under treatment paths that are never observed. This reminds me of other linear factor model approaches to counterfactual imputation, for example https://arxiv.org/abs/2006.07691. The efficacy of such methods will clearly depend on the setting.

**Strengths:**

I found the paper to be very well-motivated and well-written, and it the approach is mostly well-explained. I think the method is reasonable and the simulation results are encouraging (particularly those in Appendix C). I think this is a common and important problem in empirical work and a particularly difficult one for causal inference. I think there is a good chance that applied researchers will apply this method.

**Weaknesses:**

The notation $\Psi_0$ at the top of page 4 is never defined. I was also unclear on how the likelihood is specified, presumably this is under an assumption that the noise vector is assumed to be independent standard Gaussian?

One section in which the explanation could be sharpened is 4.1. Personally, I found the manner in which counterfactuals are formulated to be quite counter-intuitive. I would typically think of a counterfactual for some individual in terms of the path of the outcome under a counterfactual path for the treatment but with that individual's static state s, which might be interpreted as unobserved individual characteristics, and other covariates, kept fixed. Here, the counterfactual is modeled as a change in the state that is compatible with the prior factual history of treatment and the counterfactual future treatment path. Now, I think this is a reasonable way to proceed but this could use some more elaboration. In particular, I think it might help to more clearly relate this model to the large literature on linear factor models for causal inference.

I found it rather odd that treatment is modeled as some linear combination of factors with an additive noise term. In fact, treatment is binary in most cases discussed and it can only take one of a small number of paths in the factual data. Some more discussion of this would be helpful.

In the simulation results in Section 5.1, the average path of the outcomes for untreated individuals is almost completely flat. The imputed no-treatment counterfactual paths are likewise flat on average and this matches the ground-truth in the simulation. This is something of a trivial example: there are no systematic changes in outcomes over time that cannot be attributed to treatment. I found the example in Figure 4 in Appendix C in which there are richer trends at play, much more convincing and personally I would put these in the main body of the paper instead.

If I understand correctly, the simulation adheres to the modeling assumptions of the authors. I wondered whether their methods would provide a good approximation and thus an effective inference method under alternative models that do not exactly adhere to their specification. Perhaps some kind of dynamic and slightly non-linear model.

**Questions:**

More a suggestion than a question. Although this may be beyond the scope of the current work. In my view, one thing that makes Abadie's synthetic control method (and related methods) convincing is the availability of placebo tests. What I mean by this is that, in SCM, one can pretend the treatment date was say, a month before the actual date, and impute counterfactual untreated potential outcomes for the treated group over this month. Because the treated group was indeed untreated in this period, these can be compared with the ground-truth, helping to validate (or invalidate as the case may be) the efficacy of the method. It seems like such validation approaches could also be applied in the context of the method suggested here? I think this may help researchers to assess whether their particular empirical setting is suitable for the authors' approach.

---

> ### Author Response · Authors · 2024-11-25
>
> Thank you for your positive review and constructive suggestions.
>
> - Weakness
>
> [1] "The notation $\Psi_0$ is never defined. ... presumably this is under an assumption that the noise vector is assumed to be independent standard Gaussian?"
>
> => Yes, it is an independent standard Gaussian distribution. It plays an important role in determining the characteristics of the model’s performance. We have added the missing explanation from Eq. (1), and detailed information is included in Appendix B.
>
> [2] "One section in which the explanation could be sharpened is 4.1. Personally, I found the manner in which counterfactuals are formulated to be quite counter-intuitive. ...... "
>
> => Section 4.1 outlines the fundamental assumptions of counterfactual thinking, which can be viewed from two perspectives: retrospective and prospective. This paper adopts the retrospective perspective, focusing on how outcomes "would have" differed had a different treatment been applied in the past. The challenge of generating virtual patient data in clinical trials should be addressed from this retrospective viewpoint of counterfactual prediction. For example, the Synthetic Control Method proposed by A. Abadie generates synthetic control data from this perspective.
>
> In contrast, most papers in the literature approach counterfactual questions from a prospective perspective. For instance, recurrent network models such as the Counterfactual Recurrent Network by I. Bica (ICLR 2020) and the Causal Transformer by V. Melnychuk (PMLR 2022) engage with these issues from a forward-looking standpoint. They inquire about which future outcomes "will" emerge from various treatment plans given to a patient. This perspective is particularly relevant to precision medicine, where personalized treatments are tailored for individual patients in real-world situations rather than being examined retrospectively.
>
> Equation (8) precisely presents the retrospective perspective. If a patient with factual data $x_f$ had received a counterfactual treatment $a_{cf}^{(\tau, \tau+\delta)}$, a static state $s_{cf}$ would differ as a result of a conditional probabilistic process. Counterfactual data is generated based on this principle in Eq. (7). Here, time $\tau$ refers to a point in the past, not the present.
>
> You mentioned, “I would typically think of a counterfactual for some individual in terms of the path of the outcome under a counterfactual treatment, ...... and other covariates kept fixed.” In counterfactual thinking, past covariates prior to the treatment change must be fixed, as noted in Section 3.3 $x_{cf}^{(:\tau)}=x_f^{(:\tau)}$. However, covariates can vary after the treatment change $x_{cf}^{(\tau:)}\neq x_f^{(\tau:)}$. We would like to clarify the reviewer’s understanding on this matter. As succinctly described in Eq. (1), the static state encapsulates all longitudinal data pertaining to a single patient. This concept, which incorporates treatment into the entire dataset, is introduced in Eq. (2). Therefore, when there is a change in treatment, the static state also changes. It is accurate that all covariates prior to the treatment change are fixed, as reflected in Equations (10) and (11) of Section 4.2.
>
>
> [3] "It is odd that treatment is modeled as some linear combination of factors with an additive noise term. Treatment is binary in most cases discussed and it can only take one of a small number of paths in the factual data."
>
> "I wondered whether their methods would provide a good approximation and thus an effective inference method under alternative models that do not exactly adhere to their specification. Perhaps some kind of dynamic and slightly non-linear mode."
>
> => Although Eq. (1) assumes a linear model, the GMCG algorithm of Eq. (16) derived from the combination with the GMM model in Eq. (3) is fundamentally a GMM. As mentioned in Goodfellow’s 2016 book:“ A Gaussian mixture model is a universal approximator of densities, in the sense that any smooth density can be approximated with any specific nonzero amount of error by a Gaussian mixture model with enough components,” GMM, as a universal approximator, can be applied to complex and diverse data structures as the number of mixture components $K$ increases, and it has been shown to generate data of quality comparable to GANs (Please refer to the paper of 'On GANs and GMMs, NeurIPS 2018').
>
> Specifically, the GMCG algorithm outlined in Eq. (16) is not a static GMM model; instead, it features dynamic elements where parameters such as $\delta_a$, $\mathbf{N}$, $C_k$, and $m_k$ change according to the specified treatment $\mathbf{a}_\text{cf}^{(\tau:\tau+\delta)}$, time $\tau$, interval $\delta$, and other factors. This algorithm can handle not only binary treatments but also continuously varying treatment doses. In contrast, the algorithm described in the paper by Ahn & Vashist (2024) is limited to binary treatments, highlighting a significant qualitative difference between our work and theirs.

---

> > ### Comment · Reviewer_jKvu · 2024-11-25
> >
> > To be clear, what I believe could be clarified in 4.1 is the interpretation of the static state. Does it represent unobserved characteristics of individuals, or is it simply a parameter of the individual's path? It seems to me that it is the latter, and objects of interest are ultimately identified by the assumption that every possible path for the variables must have the structure in equation (1) for some $s$.
> >
> > "any smooth density can be approximated with any specific nonzero amount of error by a Gaussian mixture model with enough components"
> > Sure, but a binary variable does not have a smooth density (or any density for that matter). What is perhaps more problematic is that $\eta_a^{(t)}$ would generally have to be dependent on $W_a^{(t)}s$ in order to ensure the left hand side of equation (2) is either zero or one, and this appears to be ruled out by the model.

---

> > > ### Author Response · Authors · 2024-11-25
> > >
> > > [1] "To be clear, what I believe could be clarified in 4.1 is the interpretation of the static state. Does it represent unobserved characteristics of individuals, or is it simply a parameter of the individual's path? It seems to me that it is the latter, and objects of interest are ultimately identified by the assumption that every possible path for the variables must have the structure in equation (1) for some s."
> > >
> > > => $s$ itself is a latent variable that is not directly observed or measured but is inferred from observed variables $x^{(t)}$. Just because $s$ is not a directly observed variable does not mean it also represents unobserved characteristics of individuals. Usually, $x^{(t)}$ represents the variables of observed data, and therefore, $s$ inferred from them can be considered the observed characteristics of individuals. However, even when there is data missingness in the variables $x^{(t)}$, the EM algorithm can be used to infer both $s$ and the missing data. In such cases, it may not be entirely incorrect to refer to them as the unobserved characteristics of individuals. However, knowing $s$ enables us to get the individual's path $x^{(t)}$ by Eq. (1), so it can be more referred to as a parameter of the individual's path.
> > >
> > > [2] "any smooth density can be approximated with any specific nonzero amount of error by a Gaussian mixture model with enough components" Sure, but a binary variable does not have a smooth density (or any density for that matter). What is perhaps more problematic is that  would generally have to be dependent on in order to ensure the left hand side of equation (2) is either zero or one, and this appears to be ruled out by the model."
> > >
> > > => This is an important question. A new paper might be required to provide a more specific and complete answer. Here, however, we can explain intuitively based on examples.
> > >
> > > The results in Table 1 correspond to binary treatment, where data exists only for the treatment and control groups. In this case, each mixture component of the GMM estimates the density for only one of the two groups. Other variables, assuming they are continuous, have their covariances calculated. Since the treatment variable is 0 or 1, the covariances can become singular, but there are two solutions. One is to reduce the dimension through the $W$ matrix so the covariances, reduced by one size, do not become singular. The other is that the GMM code provided by Scikit-Learn prevents singularity by adding a small unit matrix with reg_covar=1e-06 to the covariances (Table 1 uses the latter method). Therefore, if the treatment is simply 0 or 1, the problem can be solved without complicating the structure of the noise vector.
> > >
> > > The problem is found in the tumor data type in Appendix E and F. The radiotherapy dose can vary over time and has a binary state. Ideally, there should be numerous data for each dose path, and each mixture component should be assigned to different dose paths. This would require a much larger number ($K$) of mixture components and data. However, in reality, we may not know how many different dose paths there are, and thus, how to set $K$, and each mixture component may span different dose paths. Consequently, it can only be an approximate model. As you mentioned, we could use a complex noise model, and our team had considered using a deep GMM model, but it requires more challenging research work. Nevertheless, Appendix E and F provide better results than the latest models like CRN.
> > >
> > > Not only treatment but most clinical data have discrete data representation. Binary and categorical data do not mathematically fit the basic assumptions of our current GMCG model. It also requires further research. Despite this, the Gaussian distribution model is an excellent tool. It can be neatly expressed as in Eq. (16) due to the good properties of the Gaussian distribution model. As seen in Eq. (12), the Gaussian distribution model remains Gaussian even when cut into a lower-dimensional hyperplane. There are also methods to implement the Gaussian model on a lattice, and we hope that using such a modified Gaussian model can find a suitable method for binary/categorical data.

---

### Official Review · Reviewer_9mXH · 2024-11-03

**Soundness:** 3
**Presentation:** 2
**Contribution:** 2
**Rating:** 5
**Confidence:** 3

**Summary:**

Creating synthetic control arms is challenging in crossover trials where placebo data is lost after switching to active treatment, complicating long-term efficacy and safety assessments. We propose a Gaussian mixture model that generates counterfactual data without control data, accommodating time-varying doses and comparing treatment switchers to an extended placebo group, aiding treatment evaluation and decision-making.

**Strengths:**

Completed work: The article is clearly written, and the charts used for the presentation are visually appealing.

Clear motivation: I find the problem setting of the crossover trial is quite novel and worth thinking.

Well-organized structure: The article has a clear structure which separately introduces two key problems in the model section, followed by a detailed explanation of the methods.

Sufficient experiments: The experiments are extensive and thorough, as sufficient support for method.

**Weaknesses:**

I genuinely believe this is a well-completed work that explores a very important method. However, since I am not an expert in this field, there are certain aspects that I don’t fully understand.

First, from my understanding of causality problems, it’s important to ensure three basic assumptions such as consistency, overlap, and unconfoundedness. These assumptions are crucial to establish causal problem, and I couldn’t find explicit statements or discussions around them in the paper. Further explanation or a discussion on how the proposed method addresses or bypasses these assumptions would be helpful for clarity.

Second, while I believe this work addresses a significant and valuable problem, the paper lacks a clear articulation of the novelty of the proposed method. Explicitly positioning the contributions within the landscape of related methods would make it easier to understand the unique aspects of this approach and its value compared to existing solutions.

Third, in Section 4, the authors frequently present conclusions within framed boxes. While this can be visually helpful, it consumes a considerable amount of page space, which could be better utilized. The authors may want to consider presenting these conclusions in a more compact form, such as lemmas or theorems, which would make it easier for readers to follow the logical progression without compromising on readability.

**Questions:**

In line 161, it states, “Now the problem is to estimate a set of parameters $\mathbb{Q} = [ W, \pi_k, \mu_k, \Sigma_k, \Phi_k]$. However, the paper does not provide a clear explanation of the meaning of $\Phi_k$. Could the authors clarify what this parameter represents and its role within the estimation problem? A brief explanation or reference could help in understanding its role in the proposed model.

In Section 3.3, there are subscripts “cf” and “f” used to label variables $x$. However, it’s unclear what these subscripts signify and how they differ from each other. Could the authors elaborate on the distinctions between “cf” and “f” and provide some context as to why they are used?

---

> ### Author Response · Authors · 2024-11-25
>
> Thank you for your thoughtful, kind, and detailed questions. We have incorporated some of our responses directly into our revised paper.
>
> - Weaknesses
>
> [1] "First, it’s important to ensure three basic assumptions such as consistency, overlap, and unconfoundedness. These assumptions are crucial to establish causal problem, and I couldn’t find explicit statements or discussions around them in the paper."
>
> => This paper also adopts Rubin's potential outcome framework and is based on the standard assumptions commonly found in many studies, such as consistency, overlap, and unconfoundedness. Additionally, important new assumptions are discussed in Section 4.1, where we explore the concept of counterfactual thinking. There are two types of counterfactual thinking: retrospective counterfactual thinking, which examines the effects of altering past treatments at the current time, and prospective counterfactual thinking, which forecasts the outcomes of choosing among various possible future treatments at the present moment. This paper focuses on the retrospective perspective. We will include and organize the framework and assumptions in the appendix.
>
> [2] "Second, the paper lacks a clear articulation of the novelty of the proposed method. Explicitly positioning the contributions within the landscape of related methods would make it easier to understand the unique aspects of this approach and its value compared to existing solutions."
>
> => This paper addresses new challenges that have recently arisen in clinical trials. Specifically, we propose the first-ever machine learning algorithm designed for the common real-world scenario where control data is not available for model training.
>
> The Synthetic Control Method and its derivative algorithms cannot solve the unavailable control data case problems. State-of-the-art models like Counterfactual Recurrent Network and Causal Transformer can generate counterfactual data for longitudinal data, but many of these models perform counterfactual prediction from a prospective perspective. While this may be suitable for precision medicine, it may not be effective for clinical trials, where counterfactual data can be better generated retrospectively and compared to calculate causal effects.
>
> Despite the long-recognized issues of high costs, long durations, and difficulties in patient recruitment in modern pharmaceutical clinical trials, the impact of generative AI methodologies has not yet reached the individual and specific technical problems of generating virtual patient data. Despite many approaches and models in the field of causal inference, there is yet to be a clear solution to the type of problem presented in this paper.
>
> The problem we aim to solve, and the methodology we propose is novel.
>
> - Questions
>
> [1] "In line 161, it states, “Now the problem is to estimate a set of parameters $Q=[W, \pi_k, \mu_k, \Sigma_k, \Phi]$. However, the paper does not provide a clear explanation of the meaning of $\Phi$. Could the authors clarify what this parameter represents and its role within the estimation problem?"
>
> => $\Phi$ is the covariance matrix for the noise $\eta$ that appears in Eq. (1). The noise model plays an important role. In this paper, similar to traditional factor analysis, we assume $\Phi$ to be a diagonal matrix, which determines the overall characteristics of the model.  For more details, please refer to Appendix B.
>
> [2] "In Section 3.3, there are subscripts “cf” and “f” used to label variables $x$. However, it’s unclear what these subscripts signify and how they differ from each other."
>
> => The subscript "f" refers to factual data, while "cf" is derived from the initial letters of "counterfactual." Factual data is obtained through observation, whereas counterfactual data is generated through counterfactual thinking, which involves hypothetical scenarios that have not been observed in the real world. For example, one might consider what would happen if a patient who took medication had not taken it. The goal of this paper is to obtain counterfactual data. In the context of clinical trials, this can be interpreted as virtual patient data or a synthetic placebo arm.

---

> ### Comment · Reviewer_9mXH · 2024-11-26
>
> Thanks for your response, since the authors only illustrate parts of my "Weakness" and I could not find the assumptions in the Appendix. I will keep my score.

---

> > ### Author Response · Authors · 2024-11-28
> >
> > We have revised the paper to address all the weaknesses you pointed out and have uploaded the new version. Could you please take a moment to review Appendices G and H? We have summarized the three basic assumptions using our notation and added new assumptions this paper also requires. Although we considered organizing the framed boxes in Section 4 as you suggested, the paper does not exceed 9 pages, so there are no page constraints, and we have left them as they are for now. Instead, we have provided the proof for Equation (16), which is the final expression of the algorithm proposed in the paper, in Appendix H. It is not a difficult proof, so you should be able to verify it quickly. The two framed boxes in Section 4.1 are assumptions specifically used in this paper and do not require proof, so we did not add further explanations in the Appendix. If there are any unresolved or insufficient parts, please point them out, and we will address them in future revisions. Thank you for taking the time to review our paper.

---

### Official Review · Reviewer_APKN · 2024-11-04

**Soundness:** 3
**Presentation:** 2
**Contribution:** 2
**Rating:** 5
**Confidence:** 3

**Summary:**

This paper tackles a difficulty in learning counterfactual data -- control group data may be missing or scarce. The authors propose a Gaussian mixture counterfactual generator and demonstrate it performance in numerical examples.

**Strengths:**

Introducing the Gaussian mixture counterfactual generator is relatively clear to the readers.

There are interesting numerical results demonstrating the effectiveness of the proposed method.

**Weaknesses:**

I have the impression that the methodology is not well motivated. Both the static state analysis and Gaussian mixture model jumped in abruptly. In addition, from equation (3), we do not see a clear dependency of $\mathbf{s}$ on $\mathbf{x}$ and $\mathbf{a}$. However, in Section 4, the dependence appears without discussion.

It is unclear how to find the parameters $\boldsymbol{\mu}$ and $\boldsymbol{\Sigma}$ in Gaussian mixture.

I am not convinced why does the proposed method tackle the difficult situation of scarce or no control group data.

**Questions:**

Is there a particular reason to consider the static state analysis (SSA) for counterfactual distribution learning? For causal effects, I believe there are many other methods, such as doubly-robust methods, IPTW, marginal structural models.

What is $k$ -- the number of components in Gaussian mixture -- used in experiments? How do you find the parameters in Gaussian mixture?

The kernel matrix in equation (11) is unclear to me.

---

> ### Author Response · Authors · 2024-11-25
>
> Thank you for your honest review comments on our paper. We have revised the paper to address the parts that could be immediately corrected based on your comments. Our responses to each of the questions or comments are as follows:
>
> - Weakness
>
> [1] "The methodology is not well motivated. Both SSA and GMM jumped in abruptly."
>
> => The motivation related to this is briefly described in Section 2.2. It discusses the limitations and new possibilities of the paper (Ahn \& Vashist, ICLR 2024). SSA and GMM are the methodologies used in that paper, and our paper also starts from this foundation. More specifically, the SSA model expressed in Eq. (1) is the same as that in the paper, Eq. (2) is a new introduction and our key contribution, and Eq. (3) is slightly different from the GMM of that paper. The GMM in that paper is dependent on treatment, whereas our GMM in Eq. (3) is independent of treatment. We have revised the first part of the Model section to better highlight the connection with the literature and the motivation of the paper.
>
> [2] "From Eq. (3), we do not see a clear dependency of s on x and a. However, in Section 4, the dependence appears without discussion."
>
> => Equation (3) itself does not have a dependency of $\mathbf{s}$ on $\mathbf{x}$ or $\mathbf{a}$. Eq. (3) shows that $\mathbf{s}$ follows a Gaussian mixture probability model, and the dependency of $\mathbf{s}$ on $\mathbf{x}$ or $\mathbf{a}$ is expressed in Eq. (1). Therefore, our model is constructed by combining Equations (1) and (3). Specifically, for more details, you can refer to Equations (22) or (24) in Appendix B to confirm the dependency of $\mathbf{s}$ on $\mathbf{x}$ or $\mathbf{a}$.
>
> [3] "It is unclear how to find $\mu$ and $\Sigma$."
>
> => In Appendix B, we have summarized the EM algorithms to find the values of $\mu$ and $\Sigma$.
>
> [4] "I'm not convinced why the proposed method tackles the difficult situation of no control group data."
>
> => The method proposed in this paper is the Gaussian Mixture Model (GMM) expressed in Eq. (16). By inputting the factual data $x_f$ and an alternative treatment phenotype $\delta_a$, the model can generate counterfactual data $x_{cf}$. While GMM is typically used for clustering and density estimation, its new role as a counterfactual generator is a key contribution. However, the GMCG model is not just a static GMM model. It dynamically changes variables (e.g., $\mathbf{m}_k$) based on the given treatment and intervention time. As shown in Section 5.1, assuming linear efficacy, fitting a single Gaussian distribution model to the (1mg dose - outcome) data and the (2mg dose - outcome) data allows for the prediction of the (0mg dose - outcome) data. This is a form of extrapolation. A single Gaussian distribution model can effectively model linear dose-response. However, a single Gaussian distribution model is insufficient for nonlinear dose-response, and multiple mixture models are needed. Our GMCG algorithm dynamically fits well with general and diverse treatment plans that change over time.
>
> - Questions
>
> [1] "Is there a particular reason to consider SSA for counterfactual distribution learning? For causal effects, I believe there are other methods, such as IPTW."
>
> => Our paper builds upon the work presented in the paper by Ahn \& Vashist (2024), which was published at ICLR 2024. Section 2.2 points out the limitations of that paper and suggests the possibility of transitioning to an S-learner framework. In that paper, a model called static state analysis is proposed to algebraically satisfy the causality conditions for pre-treatment in longitudinal data. Our paper also develops its content based on this foundation.
>
> While doubly-robust methods, IPTW, and marginal structural models are all excellent and rigorous causal inference models, using a latent variable factor model like SSA represents an independent research stream. For example, “The Blessings of Multiple Causes (Y. Wang, 2019)” and “Time Series Deconfounder (I. Bica, 2020)” are also good papers in this direction. In the case of SSA, as shown in Eq. (16), a concise GMM-based counterfactual generator can be obtained. In the healthcare sector, such as clinical trials, it is common to deal with small-scale data, and in such cases, efforts to generate synthetic data through small and concise models are necessary. Although the model is small, as shown in the LDL cholesterol simulation data and tumor simulation data experiments, the model's performance is superior.
>
> [2] "The kernel matrix in Eq. (11) is unclear."
>
> => We have corrected the term “kernel matrix” in Eq. (11). It does not refer to the kernel matrix as a Gram matrix, but rather to an arbitrary basis of the null space of the matrix $[\mathbf{W}^{(: \tau)};\mathbf{W}_a^{(\tau:\tau+\delta)}]$. Specifically, $[\mathbf{W}^{(:\tau)};\mathbf{W}_a^{(\tau:\tau+\delta)}]\mathbf{N}=\mathbf{O}$. While an orthonormal basis is commonly used, $\mathbf{N}$ does not need to be uniquely determined.

---

> > ### Comment · Reviewer_APKN · 2024-12-02
> >
> > Dear Authors,
> >
> > Thanks for the detailed responses to my questions and concerns.
> >
> > I would be happy to raise my score to a 5. As I am not an expert in the relevant topics, I am indicating my willingness to area chairs and other reviewers for deciding the overall merit of the paper.
> >
> > My only standing complaint is that the motivation and the particular choice of the methodology need careful elaborations. I appreciate the comparison to existing literature, yet also look forward to tailored explanations of adopting a method instead of using what has been used in other papers.

---

> > > ### Author Response · Authors · 2024-12-02
> > >
> > > If we are given the opportunity to revise our paper, we will make sure to provide additional logical explanations and reasons for the motivation and the methodology used in our work. Thank you for raising your score.

---

### Meta-Review · Area_Chair_36Ex · 2024-12-22

**Metareview:**

The paper seeks to develop synthetic controls for trials where placebo patients switch to active treatment (a crossover design) which removes placebo data. The synthetic controls are based on Gaussian mixtures that generates counterfactuals without specific control data needed for training.

The reviewers were positive or borderline negative. The borderline negative reviewers had concerns about the presentations and the assumptions, which I thought were adequately addressed in the author reply. Te comments by reviewer jKvu would be helpful in clarifying the paper and making the text more accessible to non-experts. For example,

"As I understand it, the method is essentially an imputation approach. We do not observe individuals who are never treated in the data, but by imposing a sufficiently restrictive model for the relationship between all variables including treatment, we can impute outcomes under treatment paths that are never observed."

This is a bit unlike the original synthetic controls where there is some availability of placebo.

**Additional Comments On Reviewer Discussion:**

Two of the reviewers were quite positive. The other two were borderline negative.  One of the reviewers raised their score to an accept

---

### Decision · Program_Chairs · 2025-01-22

Accept (Poster)